# AEMP: Autoregressive-Enhanced Masked Pre-training for Robust Indoor Localization

## Abstract

The major obstacle for learning-based Channel State Information (CSI) localization is to obtain a high-quality large-scale annotated dataset. However, unlike visual datasets that can be easily annotated by human workers, CSI signals are RF signal is non-intuitive and non-interpretable, making the annotation process both time-consuming and labor-intensive. Considering the potential of self-supervised learning to reduce reliance on labeled data, masked reconstruction has emerged as a promising alternative. However, directly applying existing designs to large-scale CSI scenarios faces unique challenges, including unstable representations in unmasked regions, inability to preserve long-range channel correlations, and high sensitivity to variations in access point layouts and propagation environments. To address these issues, we propose an autoregressive-enhanced masked pre-training (AEMP) framework. AEMP employs a hierarchical Transformer architecture where spatial subnetworks perform masked reconstruction to capture local channel features, while a temporal network enforces consistency through autoregressive prediction. In addition, multi-view fusion and span masking improve robustness under dynamic deployment conditions. Extensive experiments demonstrate that AEMP yields stable and transferable representations, achieving superior performance and strong generalization on downstream indoor localization tasks. To the best of our knowledge, this is the first pre-training framework for wireless sensing that integrates temporal prediction to complement masked reconstruction.

## 1 Introduction

Despite the significant progress in GPS-based outdoor navigation (Kaplan & Hegarty, 2017), its application in indoor localization is fundamentally limited by obstructed satellite signals and insufficient satellite visibility. In contrast, WiFi localization leverages existing infrastructure without the need for additional hardware deployment and provides adequate coverage for indoor environments. Existing studies have utilized various WiFi signal metrics for localization, including Received Signal Strength Indicator (RSSI) (Ni et al., 2003; Ibrahim et al., 2018), carrier phase (Yang et al., 2014; Ma et al., 2017), Time of Flight (ToF) (Mariakakis et al., 2014), Angle of Arrival (AoA) (Xie et al., 2018; An et al., 2020), Channel Impulse Response (CIR) (Gao et al., 2024) and Channel State Information (CSI) (Xie et al., 2019). Among these, CSI has emerged as a promising solution because it captures detailed attenuation and phase shift information at the granularity of Orthogonal Frequency-Division Multiplexing (OFDM) subcarriers (Yang et al., 2013). By exploiting the channel characteristics reflected in CSI, we can enable applications such as navigating to a conference room in a new building or finding a product of interest in a shopping mall, providing significant benefits for daily life.

Existing CSI-based data-driven systems (Abbas et al., 2019) are typically deployed and developed under controlled conditions, which limits their applicability to real-world scenarios, as shown in Table 1. To bridge this gap, we explore a CSI learning-driven localization system within a large-scale ISAC (Integrated Sensing and Communication) platform. This platform operates under the centralized control of a WLAN controller, simultaneously supporting daily communication needs and enabling CSI acquisition for sensing and localization.

However, real-world ISAC platforms often operate under dynamic conditions, such as varying numbers of available access points (APs) and non-line-of-sight (NLoS) environments. These factors

Table 1: Comparison of CSI-Based Learning-based Localization Methods and Our Deployment

| System | APs | AP Types | CSI Dim. | Unlab. Data |
|---|---|---|---|---|
| ConFi (Chen et al., 2017) | 3 | Intel 5300 | $3 \times 1 \times 30$ | ✗ |
| DLM (Arnold et al., 2019) | 1 | USRP | $64 \times 1 \times 922$ | ✗ |
| DLoc (Ayyalasomayajula et al., 2020) | 3 or 4 | Quantenna APs | $4 \times 1 \times 216$ | ✗ |
| RLoc (Zhang et al., 2024) | 3 or 4 | Intel 5300 | $3 \times 1 \times 30$ | ✗ |
| MSG (Liu et al., 2025) | 4 | Intel 5300 | $3 \times 1 \times 30$ | ✗ |
| RoArray (Gong & Liu, 2019) | 6 | Intel 5300 | $3 \times 1 \times 30$ | ✗ |
| OrchlLoc (Yang et al., 2024) | 64 | LoRa | $2 \times 1 \times 8$ | ✗ |
| **Our Deployment** | **13-79** | **H3C WA6520 WA6526E** | $2 \times 1 \times 42$ | ✓ |

make the collection of large-scale labeled datasets particularly challenging. This naturally raises a fundamental question: *how can we design models that generalize across diverse scenarios with limited supervision?* Recent advances (Caron et al., 2021; Li et al., 2021; Bardes et al., 2022) in pre-training paradigms offer a promising direction to address this challenge. Among these, masked modeling has emerged as a widely adopted strategy in vision and language. It learns powerful, task-agnostic representations by forcing a model to reconstruct data from a corrupted input. Yet, when applied directly to CSI signals, conventional masked modeling faces three critical limitations:

**Lack of Supervision on Unmasked Tokens.** Standard masked modeling primarily enforces reconstruction on the masked portion of the input, leaving the unmasked tokens unconstrained. For CSI signals, this leads to unstable representation learning, as unmasked tokens may drift without alignment to the underlying physical semantics, thereby weakening overall feature consistency.

**Local–Global Inconsistency.** CSI inherently encodes both local channel fluctuations and global spatial correlations across multiple APs. However, traditional masked modeling focuses on reconstructing local masked segments independently, failing to guarantee consistency with global RF propagation patterns. This mismatch undermines the ability to capture coherent spatio-temporal dependencies.

**Sensitivity to Deployment Variations.** Unlike images with relatively stable pixel statistics, CSI signals are highly sensitive to AP layouts, antenna configurations, and NLoS propagation. Random masking in these models may inadvertently discard critical components, making the learned representations brittle and less robust under changing deployment conditions.

To address the spatiotemporal dependencies of wireless signals and the limitations of masked modeling, we propose a novel hybrid pre-training paradigm called Autoregressive-Enhanced Masked Pre-training (AEMP). The framework features two core self-supervised tasks, modeling in both the spatial and temporal domains. Specifically, we design a hierarchical Transformer architecture that consists of multiple parameter-shared spatial subnetworks (encoder) and a temporal subnetwork (decoder). Due to the spatial properties of wireless signals, the spatial subnetwork performs masked reconstruction within each frame to learn local spatial features. We employ a multi-view combination strategy to reduce the reliance on specific AP combinations. In addition, we introduce a span masking mechanism (Joshi et al., 2020) to simulate dynamic deployment conditions in real-world scenarios. Beyond the masked reconstruction task handled by the spatial subnetwork, we also utilize the temporal subnetwork to perform an autoregressive prediction task, which forces the reconstruction output to maintain global consistency within its context. Finally, we use the jointly pre-trained spatial subnetwork representations as feature inputs for the fine-tuning stage to enhance CSI-based indoor localization. With our meticulously designed pre-training framework, our model outperforms state-of-the-art methods on downstream indoor localization tasks, with an average median error of 0.90 m and an average tail error of 2.65 m.

Our main contribution can be summarized as:

- We introduce the first pre-training framework that integrates autoregressive prediction with masked modeling for wireless sensing, addressing the limitations of conventional masked modeling on temporal signals.

- We design a novel hierarchical Transformer architecture with parameter-shared spatial subnetwork and a temporal subnetwork to effectively capture both local features and global temporal dependencies.
- Our AEMP framework achieves superior performance and robust generalization on indoor localization tasks, surpassing state-of-the-art methods in dynamic, real-world deployments.

## 2 RELATED WORK

**CSI-Based Localization.** CSI-based localization approaches are typically grouped into three categories, namely angle-based (Zhang et al., 2022; Tai et al., 2019; Yang et al., 2023), range-based (Vasisht et al., 2016; Zhang et al., 2020), and data-driven methods (Ayyalasomayajula et al., 2020). Angle-based techniques leverage array signal processing to infer the AoA, while range-based solutions often depend on strategies such as channel dropping. Both, however, are constrained by antenna array geometry (Chen et al., 2012) and strict communication requirements (Müller & Röhrig, 2022). Data-driven methods can be further divided into fingerprinting (Hu et al., 2022; Ni et al., 2022; Wang et al., 2017) and learning-based approaches (Ruan et al., 2022; Xu et al., 2024). Fingerprinting is a two-phase process: an offline phase builds a database that maps channel features to ground-truth positions, while an online phase matches new observations to the closest entry for localization. In contrast, learning-based approaches employ deep neural networks to directly learn an end-to-end mapping from CSI measurements to spatial coordinates, bypassing the need for a pre-built database. We propose a learning-based method for robust and accurate indoor localization. By using the CSI spatial covariance matrix as our model's input, we enable it to learn unique signal patterns directly correlated with a device's position, even in challenging multipath and NLOS environments.

**Self-Supervised Wireless Sensing.** The increasing prevalence of wireless communication systems has motivated research on extracting meaningful representations from radio signals using deep learning. Traditional wireless sensing models, however, often underperform in few-shot scenarios, making self-supervised learning (SSL) a promising alternative. Recent studies explore various learning paradigms. Transformers such as LLM4CP (Liu et al., 2024) and Trans4CP (Jiang et al., 2022) achieve accurate physical-layer channel estimation, while TMAENG (Zayat et al., 2024) employs a Transformer-based masked autoencoder to address resource allocation challenges. LLMPhy (Lee et al., 2024) leverages pre-trained language models to enhance the robustness of physical-layer communication. Other approaches focus on multi-modal and multi-task learning. LWM (Alikhani et al., 2025) utilizes large-scale wireless datasets to improve channel modeling, CSI-based DT (Jiao et al., 2024) applies contrastive learning for multi-task zero-shot learning, ChannelGPT (Yu et al., 2024) integrates multi-modal data for accurate parametric channel generation, and MMTBeam (Tian et al., 2023) fuses sensor inputs from cameras, LiDAR, and GPS to explore beam prediction. HMP-LLM (Zhong et al., 2024) is capable of performing human mobility prediction in zero-shot scenarios. Nevertheless, most methods concentrate on specific physical-layer tasks or multi-modal fusion, and their generalization to novel environments is limited. To address this, we propose AEMP, a self-supervised framework that combines masked modeling with autoregressive prediction to learn robust spatiotemporal representations, improving generalization under data scarcity.

## 3 METHOD

### 3.1 PRELIMINARY

#### 3.1.1 WIFI MOTION SENSING

In the field of wireless sensing, CSI typically describes the propagation of the signal from the transmitter (TX) to the receiver (RX). Since the transmitted CSI signal undergoes multipath effects during propagation in indoor environments, the CSI can be written as a sum of signals propagating along different paths:

$$H(t, f) = \sum_{l=1}^{L} \alpha_l(t)e^{-j2\pi f\tau_l(t)} + n(t, f), \tag{1}$$

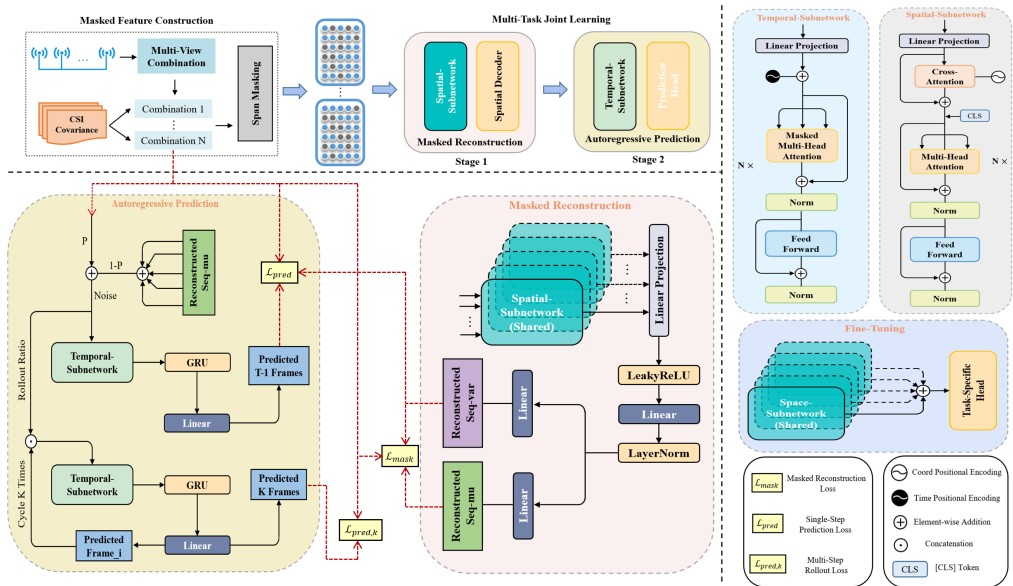

Figure 1: The overview of CSI-based indoor localization pretraining model. AEMP consists of a masked feature construction module and a multi-task joint learning module.

where $\alpha_l$ and $\tau_l$ represent the attenuation coefficient and propagation delay of the $l$-th path component, respectively, $n(t, f)$ represents additive white Gaussian noise, and $L$ is the number of paths.

The received CSI can be formulated as $\mathbf{H} \in \mathbb{C}^{N_R \times N_K}$, where $N_R$ represents the number of antennas and $N_K$ the number of subcarriers. The corresponding spatial covariance $\mathbf{C}$ can be extracted by:

$$\mathbf{C} = \frac{1}{N_K} \sum_{k=1}^{N_K} \mathbf{h}_k \mathbf{h}_k^H \tag{2}$$

where $\mathbf{h}_k \in \mathbb{C}^{N_R \times 1}$ indicate the channel vector on the $k$-th subcarrier. The real part, imaginary part, and amplitude of the covariance matrix are then concatenated to form the input features. Our AEMP leverages this sensing information, using a joint learning block to combine local spatial features and global temporal features, which enables comprehensive modeling of wireless signal representations for human activity and localization.

### 3.1.2 SPAN MASKING

As a masking strategy, Span Masking aims to more effectively capture and predict spans of tokens by masking consecutive subsequences rather than individual tokens. At each iteration, we first sample the length of the masked subsequence (denoted as $l$) from a geometric distribution $Geo(p)$, clipped at $l_{\max}$:

$$P(l = k) = (1 - p)^{k-1} p, \quad s.t. \, l \in [1, l_{\max}], \tag{3}$$

where $p$ denotes the success probability and $l_{\max}$ specifies the maximum span length. The starting position of the span is then uniformly sampled. For indoor localization systems, we extend this strategy into a two-dimensional span masking scheme, which adaptively preserves local spatial patterns. This design enables alignment with the underlying physical semantics and more faithfully reflects real-world failure modes, such as consecutive packet loss or regional signal blockage, thereby improving robustness under distribution shifts.

### 3.2 OVERALL ARCHITECTURE

As depicted in Figure 1, AEMP is a multi-stage pre-training network comprising masked feature construction and multi-task joint learning. Specifically, to enhance robustness to varying AP configurations, we perform multi-view combination augmentation and span masking strategy on the

sampled $T$ CSI spatial covariances with the shape $\mathbf{F}_c \in \mathbb{C}^{(T \times N \times F)}$. The core of the framework is a multi-task joint learning paradigm based on self-supervision, which includes masked reconstruction and autoregressive prediction. Mask reconstruction uses a BERT-style spatial subnetwork (encoder) to learn spatial representations of intra-frame AP-combination distributions and to output the reconstructed mean and log-variance, thereby quantifying the model's predictive confidence. In contrast, the autoregressive prediction task is handled by a GPT-style temporal subnetwork (decoder), which predicts the reconstructed CSI features. This decoder not only performs single-step prediction to capture short-term temporal dependencies but also incorporates a multi-step rollout mechanism to enhance its capacity for long-horizon temporal reasoning. Finally, during fine-tuning, the encoded output of the spatial subnetwork is used as a high-dimensional input for a lightweight task-specific head, which generates predictions for downstream localization.

### 3.3 MASKED FEATURE CONSTRUCTION

To better utilize spatial information, we introduce a multi-view combination strategy for data augmentation. For each time frame, we select a subset of $N-1$ APs from a pool of the top $N$ APs with the strongest RSSI, which yields $C_N^{N-1}$ different combinations. After the combination operation, the shape of $\mathbf{F}_c$ becomes $\mathbf{F}_c \in \mathbb{C}^{(T \times C_N^{N-1} \times N-1 \times F)}$, where $N = 6$. To complement the masked reconstruction task, we use a 2D span masking strategy to mask contiguous spans in the input. This process is applied to a randomly selected portion of time frames within each data sample. For the masked positions, we replace their features with zero at an 80% probability, a random value at a 10% probability, and leave them unchanged at a 10% probability.

### 3.4 MULTI-TASK JOINT LEARNING

As illustrated in Figure 1, our framework for multi-task joint learning consists of two self-supervised tasks: masked reconstruction and autoregressive prediction. The objective is to learn robust representations for indoor localization by leveraging both spatial context and temporal dynamics. The masked reconstruction module focuses on spatial feature learning, while the autoregressive prediction module handles temporal modeling. A key architectural feature of this framework is its unique error propagation mechanism, where temporal consistency errors from the autoregressive prediction are back-propagated to the masked reconstruction encoder as a regularization term. This design forces the model to learn a representation that is not only spatially coherent but also physically consistent over time. This distinct cross-task supervision ensures the model learns a predictable global frame structure, making it highly suitable for temporal reasoning tasks.

### 3.5 MASKED RECONSTRUCTION

Masked Reconstruction extracts a context-aware spatial representation from each time frame using a hierarchical Transformer architecture, where multiple spatial subnetworks with shared parameters process frames independently. For a given frame, the input data $\mathbf{F}_c$ is projected to a feature embedding $\mathbf{F}_e$, and the AP's 2D physical coordinates $\mathbf{P}_{xy}$ are encoded via a frequency-based positional embedding $\text{PE}_f$. The feature and positional embeddings are integrated through a cross-attention mechanism, formulated as:

$$\mathbf{F}_{\text{out}} = \mathbf{F}_e + \text{CrossAttention}(\mathbf{F}_e, \text{PE}_f(\mathbf{P}_{xy})), \tag{4}$$

where $\mathbf{F}_{\text{out}}$ represents the fused feature embedding. A *CLS* token is prepended to the sequence to obtain a global frame-level representation, which is then processed by a BERT-style encoder for downstream fine-tuning.

The encoded representations are passed to a spatial decoder, a module that explicitly models the inherent noise and uncertainty within ISAC data. It outputs a reconstructed mean $\mu$ and log-variance $\log \sigma$ for each position. The final objective is to optimize the Gaussian Negative Log-Likelihood (NLL) loss (Nix & Weigend, 1994), formulated as:

$$\mathcal{L}_{\text{mask}} = \frac{1}{2} \cdot \frac{1}{N_{\text{mask}}} \sum_{i=1}^{N_{\text{mask}}} \left[ \frac{(\mathbf{Y}_i - \mu_i)^2}{\exp(\log \sigma_i^2)} + \log \sigma_i^2 + \log(2\pi) \right], \tag{5}$$

where $N_{\mathrm{mask}}$ represents the number of masked tokens, while $\mathbf{Y}$ denotes the unmasked original input. This loss function not only minimizes the reconstruction error but also enables the model to predict its own confidence.

## 3.6 AUTOREGRESSIVE PREDICTION

Autoregressive prediction leverages a temporal subnetwork to capture the temporal dynamics of the data sequence. This module is trained on a single-step conditional prediction task, which is further extended with a multi-step autoregressive rollout. The input for the subnetwork is dynamically constructed using a teacher forcing scheduler that controls a probability $p$. This probability gradually decreases with each training step. At each time step $t$, the input $\mathbf{X}_t$ is determined by a weighted combination of the mean $\mu_t$ from the masked reconstruction output and the unmasked original input $\mathbf{Y}_t$, formulated as:

$$\mathbf{X}_t = (1 - b_t)\mu_t + b_t\mathbf{Y}_t, \quad b_t \sim \mathrm{Bernoulli}(p), \tag{6}$$

where $b_t$ is sampled from a Bernoulli distribution with probability $p$. The mixed input $\mathbf{X}_t$ is then augmented with noise to improve robustness. The resulting sequence is finally passed through a sinusoidal positional encoding layer to inject information about the frame sequence order.

### 3.6.1 SINGLE-STEP PREDICTION

As the primary training strategy, the output of the temporal subnetwork is processed by a GRU and a linear head to generate predictions for the next frame $\hat{\mathbf{Y}}_{t+1}$. The single-step prediction loss $\mathcal{L}_{\mathrm{pred}}$ is calculated using a weighted Mean Squared Error (MSE) (Lee, 1998), where the weight for each predicted feature is dynamically determined by the uncertainty estimated by the masked reconstruction.

$$\mathcal{L}_{\mathrm{pred}} = \mathrm{MSE}\left((\mathbf{Y}_{t+1} - \hat{\mathbf{Y}}_{t+1})^2 \odot \mathbf{W}_t\right), \quad \mathbf{W}_t = \frac{1}{1 + \xi \cdot \exp(\log \sigma_t^2)}. \tag{7}$$

The weight matrix $\mathbf{W}_t$ is derived from the reconstructed log-variance $\log \sigma$ of the previous frame and $\xi$ denotes the weighting coefficient. This weighting scheme prioritizes learning from predictions with high confidence.

### 3.6.2 MULTI-STEP ROLLOUT

To enhance the long-term forecasting capability of the model, the model is required to generate a sequence of $k$ future frames in an autoregressive manner with a low probability. The rollout begins with a brief starting sequence composed of ground truth frames. For the $i$-th step of the rollout, the prediction $\hat{\mathbf{Y}}_{t_0+i}$ is a function of the starting sequence and all prior predictions, formulated as:

$$\hat{\mathbf{Y}}_{t_0+i} = \mathcal{G}\left([\mathbf{Y}_{1..t_0}, \hat{\mathbf{Y}}_{t_0+1..t_0+i-1}]\right). \tag{8}$$

Here, $\mathcal{G}$ represents the generative process of the Transformer decoder and its task heads, and $[\cdot]$ denotes concatenation. In each step of the rollout, the model predicts the next frame, then projects and appends this prediction to the memory of the decoder to serve as context for the next prediction. The multi-step rollout loss $\mathcal{L}_{\mathrm{pred},k}$ is a simple MSE between the predicted and ground truth sequences.

The total pre-training loss $\mathcal{L}_{\mathrm{total}}$ is the weighted sum of the two task losses. The weight $\lambda$ for the autoregressive prediction loss is gradually increased over the course of training using a cosine scheduler, allowing the focus of the model to smoothly transition from spatial representation to temporal dynamics. The $\mathcal{L}_{\mathrm{total}}$ can be formulated as follows:

$$\mathcal{L}_{\mathrm{total}} = \mathcal{L}_{\mathrm{mask}} + \lambda \cdot ((1 - \eta)\mathcal{L}_{\mathrm{pred}} + \eta\mathcal{L}_{\mathrm{pred},k}). \tag{9}$$

where $\eta$ controls the relative importance of the two losses in the autoregressive prediction.

## 3.7 FINE-TUNING FRAMEWORK

To facilitate the subsequent fine-tuning task, we use the *CLS* representation, encoded by a pretrained spatial subnetwork, as the input for our downstream indoor localization head. We define a total loss function $L_{train}$ that consists of three core components: map legality penalty loss $L_{map}$, distance

penalty loss $L_{dist}$, and localization loss $L_{loc}$. For $L_{map}$, we apply a maximum penalty for points outside map boundaries and an exponential penalty for points within illegal areas, with the latter based on their distance to the nearest valid region. $L_{dist}$ is a threshold-based soft constraint that is activated only when the Euclidean distance between the predicted and ground-truth coordinates exceeds a predefined threshold. $L_{loc}$ is optimized using MSE. Therefore, the total loss function can be formulated as:

$$L_{\text{train}} = \alpha L_{map} + \beta L_{dist} + \gamma L_{loc}, \tag{10}$$

where $\alpha$, $\beta$, and $\gamma$ are the weight coefficients for each loss term. The detailed formula description is provided in Appendix A.1.

## 4 ISACLOC DATASET

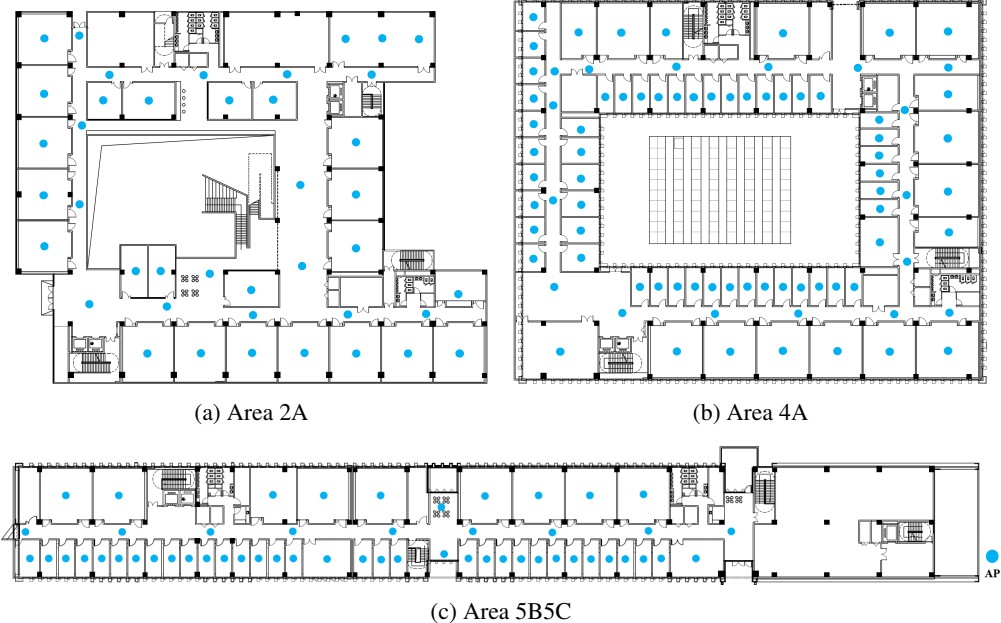

(a) Area 2A

(b) Area 4A

(c) Area 5B5C

Figure 2: Environmental scenarios of different areas. The positions of the APs are highlighted with blue dots.

While many WiFi-based indoor localization datasets exist, they are mainly confined to small-scale settings. Specifically, the unique challenges posed by large-scale ISAC platforms, including low sampling rates and signal attenuation, create a pressing need for a dedicated dataset. To fill this crucial gap, we present ISACLoc, an ISAC platform-oriented localization dataset designed to facilitate research on robust indoor positioning in realistic and complex environments.

**Data Collection.** We collect data in a large-scale real-world environment equipped with an ISAC platform, as depicted in Figure 2. The ISAC platform captures CSI data by tracking the MAC addresses of corresponding mobile phones, which act as transmitters, while the AP devices in the environment function as receivers. A Network Time Protocol (NTP) server synchronizes all devices with millisecond-level precision. Our data collection process highlights seven distinct smartphone models: iPhone 13mini, Huawei Mate10, Honor X10, Meizu 16s, Mi8SE, Pixel 2XL and Pixel 4.

To get accurate ground truth labels for human movement, we use a pedestrian dead reckoning particle filter map-aided algorithm from (Ghaoui et al., 2023). We also leverage AP locations as anchors to constrain the cumulative error of the IMU data from the mobile phone. Additionally, the time of the mobile phone and the time of the ISAC platform are synchronized using NTP.

**ISACLoc Dataset Description.** Our ISACLoc dataset comprises approximately 210,000 pairs of CSI frames, along with the corresponding receiver AP coordinates, RSSI values, channel frequencies, and ground-truth location labels. The dataset is organized into two distinct subsets: ISACLoc-R (multi-region) and ISACLoc-P (multi-device).

ISACLoc-R includes data collected from two geographically separated areas, 4A and 5B5C. The pretraining and fine-tuning samples are collected over a one-week period, while the short-term test data for both areas are gathered over two days, all within the month of July. The long-term trials span 21 days in area 4A and 31 days in area 5B5C, with corresponding test data collected starting in May, ensuring temporal diversity for robust evaluation.

ISACLoc-P consists of data captured in the 2A hall area using multiple mobile devices operated by different participants. A total of six weeks of data are collected, with the first week reserved for testing, encompassing CSI collected by various users and mobile devices. The remaining five weeks of data are used for pretraining and fine-tuning, involving seven smartphones used by a single participant. This subset is designed to emulate real-world deployment conditions and to assess the model's ability to generalize across unseen devices and users, which is a critical aspect for practical applications.

## 5 EXPERIMENTAL RESULTS

**Implementation Details.** All our baseline and AEMP models were trained on Nvidia A100 GPUs using a PyTorch implementation. Our experimental framework consisted of three main phases: pre-training, training from scratch, and fine-tuning. During the pre-training phase, we used the AdamW optimizer with a learning rate of 4e-4 and a batch size of 128 for 100 iterations. Both the spatial and temporal subnetworks used a 6-layer Transformer architecture, with each layer containing 8 attention heads and a feed-forward network (FFN) with a hidden size of 2048. The input embedding size was 512. We employed a progressive coupling pre-training mode: for the first 25 iterations, we detached the gradient propagation for the autoregressive prediction part. Once the masked reconstruction task stabilized, we enabled end-to-end pre-training by allowing gradients to flow through the entire network.

For models trained from scratch, we used a learning rate of 1e-4, a batch size of 64, and trained for 100 epochs. CosineAnnealingLR scheduler was used to dynamically adjust the learning rate based on the training iterations, with a minimum learning rate set to 1e-6. Fine-tuning also lasted for 100 epochs and employed a hierarchical freezing learning strategy. In the first 20 epochs, we froze the parameters of the spatial subnetwork and only trained the downstream task-specific head. For the subsequent 80 epochs, all layers were unfrozen: the first 5 layers of the spatial subnetwork were fine-tuned with a learning rate of 1e-5, while the last layer was fine-tuned at a higher rate of 5e-5. The batch size was kept at 64, and we used the same optimizer and scheduler as in the training-from-scratch phase.

**Baselines.** To demonstrate the superiority of the proposed AEMP framework, we compare it with various pretraining-based indoor localization baselines, including (1) RFM (Ott et al., 2024), (2) Wireless-SSL (Salihu et al., 2024), (3) LocalGPT (Zhao et al., 2024), (4) Glow (Zhang et al., 2025), (5) DDPMLoc (Ho et al., 2020) and (6) Bert-WiFi (Guo et al., 2022). RFM and Bert-WiFi each perform masked modeling on CIR and RSSI, respectively. Wireless-SSL utilizes various subcarrier transformations to extract channel features that are robust to fading. LocalGPT determines the AoA from different base stations. Glow designs a contrastive learning that combines spatial and temporal priors based on the structure of the graph. DDPMLoc employs a diffusion model to learn stable CSI representations through noise perturbation and iterative denoising. Our fine-tuning model uses a simple MLP as the task-specific head.

**Metrics.** We evaluate localization performance using two complementary metrics. For each predicted location, we compute the Euclidean distance to the ground-truth point, then generate the cumulative distribution function (CDF) of these distances. The median error corresponds to the 50% point of the CDF, reflecting typical localization accuracy, while the tail error corresponds to the 90% point, capturing rare but large deviations.

### 5.1 QUANTITATIVE EVALUATION ON INDOOR LOCALIZATION TASKS

As shown in Table 2, AEMP achieves state-of-the-art localization performance across diverse downstream settings. In the full-data scenario, AEMP attains an average median error of 0.90 m, significantly surpassing all baseline methods. Compared with Bert-WiFi and RFM, AEMP benefits from span masking, which explicitly simulates real-world AP signal blockage and thus compensates for

the shortcomings of random masking. In addition, the multi-view combination strategy effectively augments the data while preserving the spatial layout of APs, addressing the limitations observed in Wireless-SSL and LocalGPT. Finally, autoregressive prediction enables AEMP to capture the temporal dependencies that Glow and DDPMLoc fail to model, thereby improving its ability to generalize over long-term and continuous trajectories. Daily localization results for different areas are provided in Appendix A.2.

In addition to the full-data setting, Table 3 further evaluates performance under varying proportions of labeled data. AEMP consistently demonstrates superior generalization, achieving the lowest median error across all label ratios, with a notable improvement when only 10% of labeled data is available. This highlights its strong label-efficiency, enabling effective localization even in low-resource scenarios. Taken together, these results confirm that AEMP not only maintains stable advantages in long-term evaluation across different areas, but also adapts well under limited supervision, demonstrating robustness, scalability, and broad applicability to real-world indoor localization tasks.

Table 2: Quantitative Evaluation Results for Indoor Location Task. The notation '↓': lower is better.

| Method | Area | | Mean↓ |
|---|---|---|---|
| | 4A (m) | 5B5C (m) | |
| RFM | 1.30 / 3.53 | 1.26 / 4.03 | 1.28 / 3.78 |
| Wireless-SSL | 1.30 / 3.40 | 1.24 / 3.41 | 1.27 / 3.41 |
| LocalGPT | 1.75 / 4.79 | 1.93 / 5.61 | 1.84 / 5.20 |
| Glow | 1.36 / 3.71 | 1.52 / 4.93 | 1.44 / 4.32 |
| DDPMLoc | 1.21 / 3.41 | 1.37 / 4.43 | 1.29 / 3.92 |
| Bert-WiFi | 0.96 / 2.56 | 1.18 / 2.51 | 1.07 / 2.54 |
| **w/o** pre-training | 1.06 / 2.69 | 1.38 / 3.30 | 1.22 / 3.00 |
| AEMP | **0.76 / 2.27** | **1.03 / 3.03** | **0.90 / 2.65** |

## 5.2 GENERALIZATION CAPABILITY

To demonstrate the versatility of our method, we use the ISACLoc-P dataset for pre-training and then fine-tune and test on the ISACLoc-R dataset. The model achieves an average median error of 1.01 m and an average tail error of 2.79 m. Table 4 summarizes these results. Compared with fully supervised models, the proposed AEMP requires only a small amount of target-domain data to adapt to new propagation environments, yielding substantial performance improvements over baseline methods. This demonstrates that AEMP effectively generalizes to conditions different from its primary pre-training environment and highlights its strong potential for transfer learning and cross-domain generalization.

Furthermore, experiments show that the proposed method maintains consistently high accuracy across different types of mobile devices and diverse users. Detailed results for varying phone models and user identities are provided in Appendix A.3 and A.4.

The stable performance observed under a wide range of settings—including pre-training /fine-tuning within the same environment or across different environments, as well as generalization from fixed phones to unseen phones and from a single user to multiple users—highlights the versatility of our approach and its strong potential for deployment in diverse real-world scenarios.

Table 3: Quantitative Evaluation Results of pretraining methods under varying label ratios.

| Method | Labeled Data Ratio | | | | | |
|---|---|---|---|---|---|---|
| | 10% (m) | 20% (m) | 40% (m) | 60% (m) | 80% (m) | 100% (m) |
| RFM | 1.54 / 3.92 | 1.37 / 3.61 | 1.32 / 3.56 | 1.30 / 3.56 | 1.30 / 3.56 | 1.30 / 3.53 |
| Wireless-SSL | 1.52 / 3.63 | 1.40 / 3.58 | 1.39 / 3.53 | 1.36 / 3.48 | 1.34 / 3.38 | 1.30 / 3.40 |
| LocalGPT | 2.70 / 7.54 | 2.25 / 6.17 | 1.94 / 5.66 | 1.90 / 5.09 | 1.88 / 5.17 | 1.75 / 4.79 |
| Glow | 1.45 / 3.86 | 1.41 / 3.78 | 1.37 / 3.70 | 1.38 / 3.67 | 1.36 / 3.70 | 1.36 / 3.71 |
| DDPMLoc | 1.32 / 3.52 | 1.30 / 3.50 | 1.25 / 3.51 | 1.24 / 3.47 | 1.22 / 3.45 | 1.21 / 3.41 |
| Bert-WiFi | 1.28 / 3.31 | 1.10 / 3.04 | 1.07 / 2.75 | 0.99 / 2.62 | 1.01 / 2.64 | 0.96 / 2.56 |
| **w/o** pre-training | 2.00 / 5.00 | 1.63 / 4.09 | 1.50 / 3.83 | 1.20 / 3.04 | 1.08 / 2.81 | 1.06 / 2.69 |
| AEMP | **1.27 / 3.30** | **1.03 / 3.03** | **0.92 / 2.65** | **0.88 / 2.45** | **0.82 / 2.32** | **0.76 / 2.27** |

Table 4: Cross-area localization performance under varying labeled data ratios.

| Area | Labeled Data Ratio | | | | | |
|------|---------|---------|---------|---------|---------|---------|
| | 10% (m) | 20% (m) | 40% (m) | 60% (m) | 80% (m) | 100% (m) |
| 4A | 1.95 / 5.10 | 1.39 / 3.85 | 1.16 / 3.28 | 1.08 / 3.00 | 0.98 / 2.78 | 0.92 / 2.69 |
| 4A (**w/o** pre-training) | 2.00 / 5.00 | 1.63 / 4.09 | 1.50 / 3.83 | 1.20 / 3.04 | 1.08 / 2.81 | 1.06 / 2.69 |
| 5B5C | 3.66 / 10.86 | 2.24 / 6.79 | 1.48 / 4.42 | 1.25 / 3.37 | 1.21 / 3.20 | 1.09 / 2.89 |
| 5B5C (**w/o** pre-training) | 4.28 / 10.56 | 3.66 / 9.17 | 2.26 / 5.36 | 1.94 / 4.69 | 1.55 / 3.55 | 1.38 / 3.30 |

## 5.3 ABLATION STUDY

We use data from the 4A area of the ISACLoc-R dataset for our ablation study. This evaluates the effectiveness of two key components within the AMEP pre-training framework. Table 5 summarizes the results at different proportions of labeled data.

**Masked Reconstruction Task.** This is a core component of AEMP. Removing this task leads to a significant degradation in indoor localization performance, with the median error increasing by 32.89%. Furthermore, when fine-tuning on limited labeled data, the error also increases substantially. This highlights that learning local spatial features through reconstruction is crucial for enhancing model performance.

**Autoregressive Prediction Task.** This task enhances the pre-training process by performing autoregressive prediction on the reconstructed data from a temporal consistency perspective. Its removal leads to an increase in the localization median error of 11.84%. This demonstrates that the task plays a key role in ensuring global temporal coherence and improving model generalization.

**Span Masking.** The masking mechanism simulates realistic dynamic-deployment conditions. We replace the span mask with a random mask on top of the w/o AP model, the median localization error increases by 43.42%, indicating that span masking better encourages the model to learn deployment-invariant representations.

**Multi-View Combination.** This strategy reduces reliance on specific AP combinations. Removing the multi-view ensemble from the w/o SM model reduces the input feature dimension and increases the median localization error by 55.26%. Although the multi-view combination does not introduce additional features, it decouples channel characteristics from a fixed AP layout, thereby improving robustness across heterogeneous environments.

Table 5: Ablation studies and analysis. The w/o indicates "without". MR is the masked reconstruction task. AP is the autoregressive prediction task. SM is the span masking. MVC is the Multi-View Combination.

| Method | Labeled Data Ratio | | | | | |
|--------|---------|---------|---------|---------|---------|---------|
| | 10% (m)↓ | 20% (m)↓ | 40% (m)↓ | 60% (m)↓ | 80% (m)↓ | 100% (m)↓ |
| AEMP | **1.27** | **1.03** | **0.92** | **0.88** | **0.82** | **0.76** |
| **w/o** AP | +7.87% (1.37) | +10.68% (1.14) | +8.70% (1.00) | +13.64% (1.00) | +9.76% (0.90) | +11.84% (0.85) |
| **w/o** MR | +64.57% (2.09) | +56.31% (1.61) | +45.65% (1.34) | +34.09% (1.18) | +30.49% (1.07) | +32.89% (1.01) |
| **w/o** SM | +39.37% (1.77) | +40.78% (1.45) | +33.70% (1.23) | +32.95% (1.17) | +36.59% (1.12) | +43.42% (1.09) |
| **w/o** MVC | +41.73% (1.80) | +47.57% (1.52) | +51.09% (1.39) | +55.68% (1.37) | +56.10% (1.28) | +55.26% (1.18) |

## 6 CONCLUSION

This paper introduces AEMP, a novel framework for robust wireless indoor localization. Our approach employs a dual-task self-supervised learning paradigm that unifies masked reconstruction and autoregressive prediction within a hierarchical Transformer architecture. This pre-training strategy effectively addresses the reliance on large labeled datasets and significantly enhances the model's generalization in dynamic, real-world scenarios. The superior performance of AEMP establishes a new state-of-the-art and unlocks new possibilities for indoor localization applications within ISAC platforms.

## ETHICS STATEMENT

This work adheres to the ICLR Code of Ethics.[1] Our study involves only self-collected indoor wireless sensing data and does not include any human subjects or personally identifiable information beyond the environmental sensing context. All experimental procedures were conducted in accordance with standard safety and privacy guidelines. We have carefully considered potential risks of misuse, bias, or fairness issues, and believe that our contributions do not pose foreseeable harm. No conflicts of interest or ethical concerns beyond standard research integrity were identified.

## REPRODUCIBILITY STATEMENT

We have made every effort to ensure the reproducibility of our results. A detailed description of the experimental setup, data collection procedures, and model architectures is provided in the main text and appendix. All data used in our experiments were self-collected and are documented in detail to allow replication of the study. Parts of the source code required to reproduce our experiments will be provided in the supplementary material. Training procedures, hyperparameters, and evaluation protocols are fully described to enable reproducibility by other researchers.

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

# A   APPENDIX

## LLM USAGE STATEMENT

Large Language Models (LLMs) were used solely as a general-purpose assistive tool for translating and refining the English text of this paper. The LLM did not contribute to the research ideation, experimental design, data analysis, or interpretation of results. All scientific content, results, and conclusions are the sole work of the authors.

In this appendix, we provide further details and analysis to supplement the main findings on our AEMP framework. The content is organized as follows:

- Section A.1: Details the mathematical formulations of the soft penalty loss used in fine-tuning, including both the map legality and distance-based components.
- Section A.2: Reports long-term evaluation results in 4A and 5B5C areas, demonstrating the temporal generalization and stability of AEMP over extended testing periods.
- Section A.3: Evaluates the performance of AEMP on different types of mobile phones, including iPhone, Xiaomi, and other devices.
- Section A.4: Evaluates the performance of AEMP on different users.
- Section A.5: Provides a detailed analysis of downstream localization performance under various pre-training modes, comparing results with different proportions of labeled data.
- Section A.6: Presents qualitative evaluation results, showing a comparison of trajectory reconstruction across different methods.
- Section A.7: Uses three different techniques for fine-tuning, highlighting the performance changes of AEMP under various fine-tuning strategies.
- Section A.8: Evaluates different models on public datasets under varying proportions of labeled data to verify the generalization ability of the proposed pre-training method.
- Section A.9: Lists the actual deployment in real-world environments (hallways and corridors).

## A.1   SOFT PENALTY FOR SPATIAL AND DISTANCE CONSTRAINTS

The soft penalty loss is designed to impose both spatial legality and distance-based constraints during fine-tuning. Specifically, the loss first enforces a map legality penalty $L_{\text{map}}$. For a predicted coordinate $\hat{\mathbf{p}} = (x, y)$, if it falls outside the map boundary, a maximum penalty is assigned:

$$L_{\text{out}}(x, y) = P_{\max}. \tag{11}$$

We denote $M(x, y)$ as the binarized map, where $M(x, y) = 1$ indicates a legal area (accessible space) and $M(x, y) = 0$ represents illegal regions such as walls, columns, or other obstacles. If the prediction lies within the legal area ($M(x, y) = 1$), no penalty is applied:

$$L_{\text{legal}}(x, y) = 0, \tag{12}$$

while predictions in illegal regions ($M(x, y) = 0$) incur an exponential penalty based on their distance $d(x, y)$ to the nearest legal boundary:

$$L_{\text{illegal}}(x, y) = \exp(\alpha \cdot d(x, y)) - 1. \tag{13}$$

In addition, a distance penalty is introduced to encourage accurate localization. For each prediction, the Euclidean error relative to the ground truth $\mathbf{p}$ is formulated as:

$$e = \|\hat{\mathbf{p}} - \mathbf{p}\|_2. \tag{14}$$

When this error exceeds a threshold $\delta$, a linear penalty proportional to the excess distance is applied:

$$L_{\text{dist}}(e) = \max(0, e - \delta). \tag{15}$$

Finally, the total penalty combines the two components:

$$L = L_{\text{map}}(x, y) + \Lambda \cdot L_{\text{dist}}(e), \tag{16}$$

where $\Lambda$ controls the balance between spatial and distance penalties. This design ensures that the model not only respects the spatial constraints imposed by the environment, but also achieves accurate trajectory reconstruction.

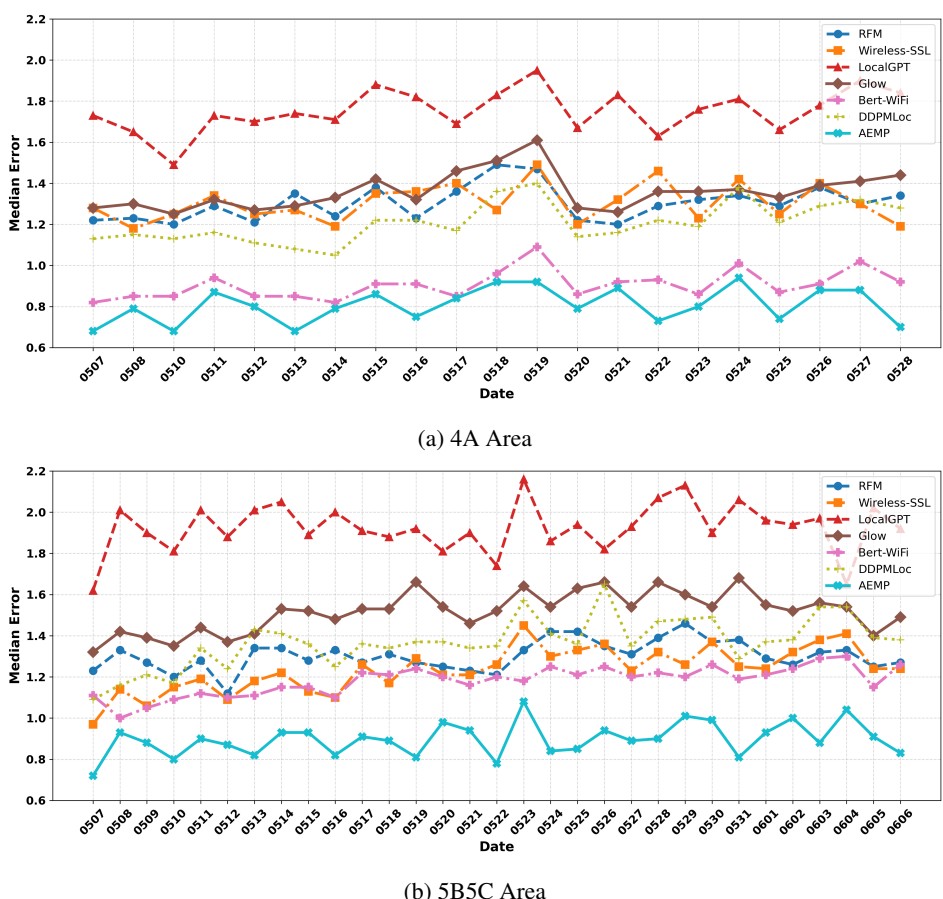

(a) 4A Area

(b) 5B5C Area

Figure 3: Daily median localization errors over long-term evaluation periods for different methods.

## A.2 LONG-TERM TEMPORAL GENERALIZATION

We further provide a detailed analysis of long-term localization performance of the AEMP-pretrained model across different areas, offering a more comprehensive demonstration of its temporal generalization ability. Figure 3 presents the test results of various methods over 21 consecutive days in 4A area (from May 7 to May 28, excluding May 9) and 31 consecutive days in 5B5C area (from May 7 to June 6).

The results show that, compared to existing methods, AEMP consistently achieves lower median errors across all areas and time periods. In particular, AEMP exhibits significantly smaller error fluctuations in long-term evaluations, demonstrating more stable and robust performance. This performance indicates that the AEMP-pretrained model maintains reliable localization accuracy even under varying dates, environmental changes, or signal fluctuations, highlighting its superior temporal generalization capability.

## A.3 CROSS-DEVICE PERFORMANCE EVALUATION

To comprehensively evaluate the performance of AEMP on different mobile phone types, we compare it against the best baseline and a non-pre-trained initial model. The dataset is sourced from ISACLoc-P, and the devices used for both pre-training and fine-tuning include iPhone 13mini, Huawei Mate10, Honor X10, Meizu 16s, Mi8SE, Pixel2XL, and Pixel 4. The tested phones include both existing and unseen devices, and the data for each was collected by different volunteers to ensure generalization across various human body characteristics.

Table 6: Indoor localization errors (m) on various iPhone devices.

| Method | Different iPhone Devices | | | | | | |
|---|---|---|---|---|---|---|---|
| | iPhone 11 | iPhone 12 | iPhone 13 | iPhone 13mini | iPhone 14 | iPhone 14pro | iPhone 15pm |
| Best Baseline | 0.85 / 2.06 | 0.97 / 2.59 | 0.90 / 2.23 | 1.87 / 4.60 | 1.19 / 3.21 | 0.84 / 2.07 | 1.11 / 3.43 |
| w/o pre-training | 1.09 / 3.16 | 1.04 / 2.46 | 1.01 / 2.71 | 2.06 / 4.74 | 1.32 / 3.62 | 1.03 / 2.45 | 1.17 / 3.81 |
| AEMP | **0.77 / 2.33** | **0.95 / 2.24** | **0.85 / 2.17** | **1.73 / 4.61** | **1.07 / 3.42** | **0.80 / 1.88** | **0.97 / 3.15** |

Table 7: Indoor localization errors (m) on various Xiaomi devices.

| Method | Different Xiaomi Devices | | | | | | |
|---|---|---|---|---|---|---|---|
| | RedmiK20 | RedmiK30 | RedmiK40 | Redminote12 | Mi8SE | Mi13-1 | Mi13-2 |
| Best Baseline | **1.60 / 5.22** | **1.55 / 11.56** | 1.01 / 2.22 | 1.05 / 2.46 | 1.11 / 3.10 | 1.10 / 3.01 | **1.43 / 5.03** |
| w/o pre-training | 1.73 / 5.87 | 1.71 / 12.36 | **0.73 / 2.73** | 1.03 / 2.55 | 0.98 / 2.56 | 1.12 / 3.12 | 1.78 / 4.03 |
| AEMP | 1.60 / 5.56 | 1.66 / 12.96 | 0.80 / 2.57 | **1.01 / 2.32** | **0.88 / 2.18** | **1.04 / 2.99** | 2.08 / 3.84 |

Table 8: Indoor localization errors (m) on other mobile devices.

| Method | Different Other Devices | | | | | |
|---|---|---|---|---|---|---|
| | 1+ace2 | Honor 60 | Honor X10 | Huawei Mate10 | Meizu 16s | Pixel 4 |
| Best Baseline | 0.89 / 2.73 | 0.87 / 2.32 | 1.18 / 4.25 | 0.86 / 2.17 | 1.02 / 2.80 | 1.66 / 4.22 |
| w/o pre-training | 1.10 / 2.76 | 0.85 / 2.33 | 1.23 / 4.17 | 0.95 / 2.22 | 1.06 / 2.64 | 1.72 / 4.45 |
| AEMP | **0.82 / 2.80** | **0.82 / 1.94** | **1.13 / 3.46** | **0.83 / 1.85** | **0.96 / 2.56** | **1.56 / 4.33** |

Tables 6, 7 and 8 report the indoor localization errors across iPhone devices, Xiaomi devices, and other mobile phones, respectively. Although the median and tail errors slightly increase compared to in-domain evaluation, the model pretrained with AEMP remains highly competitive, achieving superior accuracy on all devices except for a few Xiaomi models. These cross-device results further demonstrate the robustness and generalization capability of AEMP, which arises from its hybrid pre-training design: the spatial subnetwork with masked reconstruction and multi-view fusion reduces sensitivity to specific AP combinations and deployment variations, while the temporal autoregressive subnetwork enforces global contextual consistency. Together, these components enable AEMP to learn device-agnostic and transferable feature representations, facilitating effective knowledge transfer across heterogeneous hardware platforms and diverse user characteristics.

## A.4 CROSS-USER PERFORMANCE EVALUATION

To evaluate localization performance under realistic user diversity, we recruit 11 volunteers and instruct them to walk with the same handheld device within the designated area. Table 9 summarizes their distributions of height and weight. Each participant exhibits distinct movement patterns and physical characteristics. The results show that AEMP delivers remarkably consistent performance across users, achieving an average median error of approximately 0.81 m and an average tail error of roughly 2.27 m. This level of robustness indicates that the system effectively extracts user-independent localization features, which is a key criterion for assessing its practicality in real-world deployment.

## A.5 IMPACT OF PRE-TRAINING STRATEGIES ON LOCALIZATION PERFORMANCE

Given that our pre-training framework incorporates a two-stage multi-task joint learning module, we further conduct a detailed investigation of AEMP under different pre-training strategies. Table 10 summarizes the comparative results on indoor localization across three modes: end-to-end coupling, stop-gradient coupling, and progressive coupling. The evaluation is performed on the 4A region of the ISACLoc-R dataset, with varying proportions of labeled data to assess the effectiveness of each strategy under different levels of supervision.

Table 9: Cross-user localization performance and height-weight distribution.

| Person | 1 | 2 | 3 | 4 | 5 | 6 | 7 | 8 | 9 | 10 | 11 | Mean |
|--------|-----|-----|-----|-----|-----|-----|-----|-----|-----|-----|-----|--------|
| Height (cm) | 178 | 172 | 178 | 178 | 177 | 170 | 173 | 175 | 173 | 183 | 176 | 175.73 |
| Weight (kg) | 68 | 63 | 59 | 52 | 67 | 50 | 60 | 68 | 65 | 73 | 76 | 63.73 |
| 50th Error (m) | 0.65 | 0.72 | 0.88 | 0.91 | 0.73 | 0.75 | 0.78 | 0.92 | 0.90 | 0.85 | 0.85 | 0.81 |
| 90th Error (m) | 1.38 | 1.65 | 3.31 | 3.47 | 1.88 | 1.81 | 1.88 | 3.03 | 1.93 | 2.64 | 1.97 | 2.27 |

The results show that, compared with the other two pre-training modes, the progressively coupled scheme consistently achieves lower median localization errors. In particular, under limited labeled data, AEMP with progressive coupling maintains relatively low localization errors. This is because in the fully end-to-end mode, noisy early-stage reconstructions tend to cause overfitting, forcing the masked reconstruction task to compromise its spatial modeling capacity in order to align with autoregressive prediction.

In contrast, the stop-gradient mode reduces autoregression to a post-hoc evaluator, which cannot propagate global temporal consistency constraints back to the masked reconstruction. The progressive coupling strategy first applies stop-gradient until the masked modeling stabilizes, and then switches to end-to-end training, effectively balancing spatial and temporal modeling and thereby improving performance under limited supervised data.

Table 10: Comparison of AEMP performance under different pre-training coupling strategies. E2E is the end-to-end coupling. SG is the stop-gradient coupling. PC is the progressive coupling.

| Mode | Labeled Data Ratio | | | | | |
|------|-----------|-----------|-----------|-----------|-----------|------------|
| | 10% (m) | 20% (m) | 40% (m) | 60% (m) | 80% (m) | 100% (m) |
| E2E | 1.55 / 3.99 | 1.14 / 3.21 | 0.97 / 2.72 | 0.90 / 2.57 | 0.87 / 2.48 | 0.84 / 2.36 |
| SG | 1.58 / 4.25 | 1.13 / 3.19 | 1.03 / 2.82 | 0.88 / 2.56 | 0.85 / 2.42 | 0.82 / 2.43 |
| PC | **1.27 / 3.30** | **1.03 / 3.03** | **0.92 / 2.65** | **0.88 / 2.45** | **0.82 / 2.32** | **0.76 / 2.27** |

## A.6 QUALITATIVE ANALYSIS OF TRAJECTORY RECONSTRUCTION

Figure 4 presents a visual comparison of trajectory reconstruction results using the AEMP-pretrained model and baseline methods, complementing the quantitative evaluation. These examples illustrate the improved accuracy of AEMP in reconstructing trajectories. Although Bert-WiFi achieves overall accuracy comparable to AEMP, it exhibits noticeable deviations at certain points. RFM, which uses random mask learning for signal reconstruction, fails to produce smooth motion trajectories. Glow, constrained by its simple network architecture and graph-based input, struggles to handle NLOS corridor scenarios, resulting in substantial errors at corner regions. Wireless-SSL, which primarily focuses on subcarrier-level information while neglecting the spatial layout of APs, can only capture coarse trajectory outlines rather than precise location coordinates. These visual results highlight AEMP's ability to capture fine-grained spatial information.

## A.7 ANALYSIS OF FINE-TUNING APPROACHES IN INDOOR LOCALIZATION

Table 11 presents a comparison of localization performance across three fine-tuning approaches, including: (1) Full fine-tuning, which updates all parameters of the pretrained model on the target dataset for comprehensive adaptation; (2) LoRA, which employs a low-rank adaptation strategy by inserting lightweight trainable modules into each layer while keeping most parameters frozen, thereby recalibrating pretrained knowledge without overwriting all weights; and (3) Layer-wise fine-tuning, which progressively unfreezes and adapts parameters from higher to lower layers, making it effective when features at different depths contribute unequally to the task. All fine-tuning and evaluation are conducted on the 4A region of the ISACLoc-R dataset.

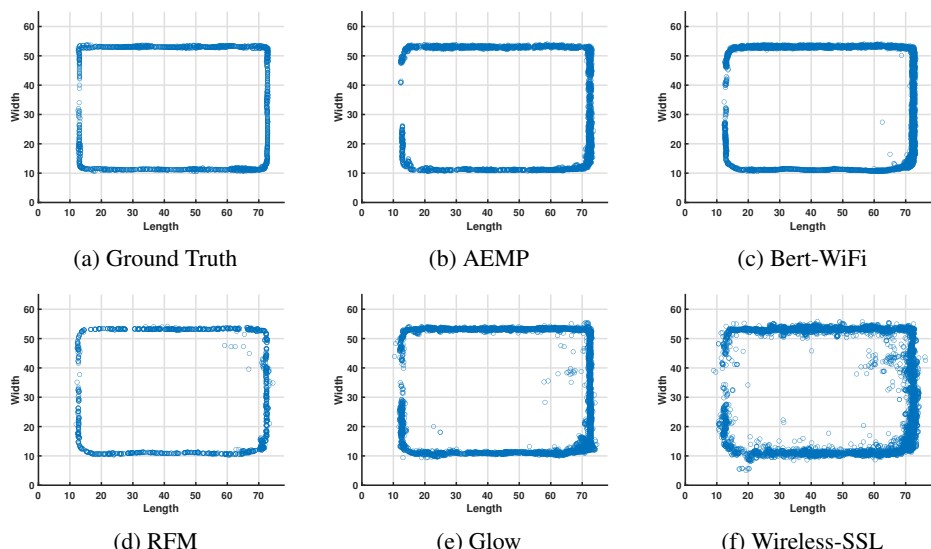

Figure 4: Qualitative comparison of trajectory reconstruction results: (a) Ground truth trajectory and reconstructed trajectories by (b) AEMP, (c) Bert-WiFi, (d) RFM, (e) Glow, and (f) Wireless-SSL.

The results show that layer-wise fine-tuning consistently achieves the lowest median and tail localization errors across different labeled data ratios. This superior performance arises from its progressive adaptation strategy: by gradually unfreezing the network from higher to lower layers, the model preserves the stability of pretrained representations in early stages while allowing deeper layers to adjust more flexibly to the downstream task. In contrast, full fine-tuning may lead to overfitting when labeled data are limited since all parameters are updated simultaneously, and LoRA, although parameter-efficient, provides weaker adaptation capacity as only low-rank modules are optimized. The layer-wise strategy thus strikes an effective balance between retaining pretrained knowledge and incorporating task-specific information, leading to consistently better localization accuracy.

Table 11: Indoor localization errors (m) under different fine-tuning approaches.

| Fine-tuning | Labeled Data Ratio | | | | | |
|---|---|---|---|---|---|---|
| | 10% (m) | 20% (m) | 40% (m) | 60% (m) | 80% (m) | 100% (m) |
| Full | 1.48 / 3.75 | 1.18 / 3.26 | 0.96 / 2.65 | 0.94 / 2.52 | 0.86 / 2.42 | 0.82 / 2.31 |
| LoRA | 2.52 / 5.80 | 1.87 / 4.60 | 1.39 / 3.48 | 1.26 / 3.16 | 1.20 / 2.91 | 1.11 / 2.79 |
| Layer-wise | **1.27 / 3.30** | **1.03 / 3.03** | **0.92 / 2.65** | **0.88 / 2.45** | **0.82 / 2.32** | **0.76 / 2.27** |

## A.8 GENERALIZATION EVALUATION ON PUBLIC DATASETS

We conduct experiments on the available WILD-v2 dataset published by DLoc (Ayyalasomayajula et al., 2020), which is collected in indoor environments and is suitable for evaluating environmental adaptation capabilities. The dataset consists of CSI samples from two visually similar environments, Env-1 and Env-2, each measuring 40mx20m size, with 6 APs, each equipped with 4 antennas. During testing, 4000 samples from Env-1 and 1000 samples from Env-2 are used as the test set. However, we only utilize Env-1 for pretraining and fine-tuning.

Table 12 presents the positioning performance of existing pretrained methods under different labeled data proportions. The results demonstrate that most existing methods perform poorly on the public dataset. This limitation is primarily due to the insufficient amount of pretraining data, which is collected within a single day and thus fails to capture time-invariant and generalizable channel features. When the model encounters unseen environments during testing, the lack of robust CSI representation leads to significant performance degradation. In contrast, the proposed method effectively

Table 12: Comparison of Localization Performance of Different Models on the WILD-v2 Dataset.

| Method | Labeled Data Ratio | | | | | |
|---|---|---|---|---|---|---|
| | 10% (m) | 20% (m) | 40% (m) | 60% (m) | 80% (m) | 100% (m) |
| RFM | 2.34 / 5.22 | 1.92 / 4.29 | 1.50 / 3.34 | 1.39 / 3.14 | 1.38 / 3.05 | 1.31 / 2.89 |
| Wireless-SSL | 2.28 / 4.77 | 2.10 / 4.34 | 1.86 / 3.92 | 1.89 / 3.80 | 1.85 / 3.71 | 1.80 / 3.68 |
| LocalGPT | 1.52 / 3.29 | 1.44 / 3.16 | 1.31 / 2.91 | 1.29 / 2.91 | 1.20 / 2.83 | 1.22 / 2.82 |
| Glow | 1.50 / 2.92 | 1.32 / 2.82 | 1.36 / 2.96 | 1.33 / 2.86 | 1.25 / 2.85 | 1.24 / 2.86 |
| Bert-WiFi | 3.07 / 5.32 | 3.08 / 5.39 | 3.12 / 5.42 | 3.09 / 5.37 | 3.03 / 5.31 | 3.04 / 5.29 |
| **AEMP** | **1.08 / 2.35** | **1.02 / 2.24** | **0.95 / 2.06** | **1.00 / 2.25** | **0.96 / 2.19** | **0.92 / 2.12** |

suppresses environment-specific interference and focuses on learning inherent location-dependent features. Consequently, even with a limited amount of single-day data for pretraining, it achieves more robust positioning performance.

### A.9 DEPLOYMENT DESCRIPTIONS IN REAL-WORLD ENVIRONMENTS

To comprehensively evaluate the performance of our work, we collect data in large-scale real-world environments equipped with the ISAC platform, including halls (2A, 4A) and corridors (5B5C). These scenarios involve varying numbers of APs and spatial sizes. Table 13 provides deployment details and behavior descriptions for each area.

Table 13: Scenario sets description.

| Area | Scenarios | Space Size | AP Number | Description |
|---|---|---|---|---|
| 2A | Central Hall | 82m × 64m | 38 | Move freely in the hall rest area. |
| 4A | Circular Hall | 79m × 65m | 79 | Walk clockwise in a square pattern along the hall corridor. |
| 5B5C | Long Corridor | 165m × 20m | 51 | Walk straight along the long corridor. |

