# OpenReview forum: "AEMP: Autoregressive-Enhanced Masked Pre-training for Robust Indoor Localization"
_ICLR.cc/2026/Conference — ICLR 2026 Conference Withdrawn Submission_

### Official Review · Reviewer_3fbY · 2025-10-30

**Soundness:** 3
**Presentation:** 3
**Contribution:** 2
**Rating:** 4
**Confidence:** 3

**Summary:**

This paper addresses the issue that traditional masked pre-training methods are unsuitable for WiFi CSI indoor localization signals, pointing out that these methods produce unstable representations and fail to capture global spatio-temporal correlations, making models highly sensitive to real-world deployment changes . To solve this, the paper proposes a novel hybrid framework called AEMP, which utilizes a spatial subnetwork for Masked Reconstruction while using a temporal subnetwork for Autoregressive Prediction. AEMP achieves state-of-the-art localization performance  and demonstrates strong generalization, especially in low-data scenarios. Additionally, the authors contribute a new large-scale real-world dataset called ISACLoc for this research.

**Strengths:**

This paper addresses the issue that traditional masked pre-training methods are unsuitable for WiFi CSI indoor localization signals, pointing out that these methods produce unstable representations and fail to capture global spatio-temporal correlations, making models highly sensitive to real-world deployment changes . To solve this, the paper proposes a novel hybrid framework called AEMP, which utilizes a spatial subnetwork for Masked Reconstruction while using a temporal subnetwork for Autoregressive Prediction  . AEMP achieves state-of-the-art localization performance and demonstrates strong generalization, especially in low-data scenarios. Additionally, the authors contribute a new large-scale real-world dataset called ISACLoc for this research.

**Weaknesses:**

“We employ a multi-view fusion strategy to reduce the reliance on specific AP combinations. In addition, we introduce a span masking mechanism (Joshi et al., 2020) to simulate dynamic deployment conditions in real-world scenarios.”
I dont see the ablation study  on these two.
Besides, I'm not sure the novelty of the MR and AP combination.

**Questions:**

Is the MR+AP combination really that useful? In other words, in your specific scenario, I feel the distinction between the two is not that significant. Also, I suggest experimenting on one or two public datasets to demonstrate the generalization performance.

---

> ### Author Response · Authors · 2025-11-21
>
> Thank you for the valuable suggestions and insights, which have significantly improved our manuscript.
>
> ---
>
> **Q1 The ablation study of the span masking mechanism and multi-view combine strategy.**
>
> Thank you for your insightful suggestion.  We apologize for the lack of clarity in the original submission and commit to revising Section 5.3 (Ablation Study) with clearer presentation and detailed analysis. To address this issue, we replaced the span masking in the w/o AP model with random masking, denoted as w/o SM. In addition, to further analyze the effectiveness of the multi-view combinations strategy, we removed the multi-view combinations based on the w/o SM model, denoted as w/o MVC. The results are summarized in the table below:
>
> |             | 10\% (m)        | 20\% (m)        | 40\% (m)        | 60\% (m)        | 80\% (m)        | 100\% (m)       |
> |-------------|-----------------|-----------------|-----------------|-----------------|-----------------|-----------------|
> | **AEMP**    | **1.27**        | **1.03**        | **0.92**        | **0.88**        | **0.82**        | **0.76**        |
> | **w/o** AP  | +7.87\% (1.37)  | +10.68\% (1.14) | +8.70\% (1.00)  | +13.64\% (1.00) | +9.76\% (0.90)  | +11.84\% (0.85) |
> | **w/o** SM  | +39.37\% (1.77) | +40.78\% (1.45) | +33.70\% (1.23) | +32.95\% (1.17) | +36.59\% (1.12) | +43.42\% (1.09) |
> | **w/o** MVC | +41.73\% (1.80) | +47.57\% (1.52) | +51.09\% (1.39) | +55.68\% (1.37) | +56.10\% (1.28) | +55.26\% (1.18) |
>
> The results show that span masking and multi-view combinations effectively improve the robustness of downstream localization and reduce both median and tail errors. This improvement arises because span masking simulates realistic dynamic deployment conditions, and the multi-view combination strategy reduces the model’s reliance on specific AP groupings, thereby encouraging the learning of deployment-invariant representations.
>
> ---
>
> **Q2 The novelty and utility of MR+AP combination.**
>
> Thank you for your valuable feedback. We greatly appreciate your insightful attention to the MR and AP combination mechanism. Your questions regarding the novelty and discriminative ability of this design in CSI scenarios directly touch upon the essence of the AEMP architecture. They motivate us to provide a clearer and more precise explanation that highlights our core innovations.
>
> - **Novelty:**
>
> Although MR and AP are known self-supervised techniques, we are the first to integrate them into a decoupled–coupled architecture for robust CSI representation learning. The novelty of our design is reflected in the following aspects:
>
> 1. Indoor CSI data inherently forms a high-dimensional sequence with strong spatiotemporal coupling. Our hierarchical Transformer explicitly assigns the MR task to the spatial sub-network, which processes high-dimensional spatial covariance features, and assigns the AP task to the temporal sub-network, which handles sequential prediction. This integrated design is the first to treat spatial completeness and temporal consistency as complementary supervisory signals, enabling synergistic feature learning that addresses both the sensitivity of traditional methods to local noise and their inability to capture long-range dependencies.
>
> 2. The cross-task regularization mechanism serves as our core innovation. In conventional MR applied to sequential data, unmasked tokens lack supervision and therefore tend to produce unstable representations. The AP loss introduces a temporal constraint that backpropagates into the MR spatial encoder, forcing local spatial representations to maintain temporal predictability. This mechanism effectively stabilizes the features learned by MR and resolves the longstanding limitation that conventional MR struggles to capture long-term temporal correlations.
>
> - **Utility:**
> Our ablation study demonstrates that both tasks are complementary and indispensable for SOTA performance. As shown in the following table, MR ensures robust spatial semantics, while AP preserves the temporal stability and predictability of these semantic features. The two components reinforce each other functionally and together form the foundation of AEMP's strong generalization ability. Their discriminative contribution to performance is both significant and essential.
>
>     |            | 10\% (m)        | 20\% (m)        | 40\% (m)        | 60\% (m)        | 80\% (m)        | 100\% (m)       |
>     |------------|-----------------|-----------------|-----------------|-----------------|-----------------|-----------------|
>     | **AEMP**   | **1.27**        | **1.03**        | **0.92**        | **0.88**        | **0.82**        | **0.76**        |
>     | **w/o** AP | +7.87\% (1.37)  | +10.68\% (1.14) | +8.70\% (1.00)  | +13.64\% (1.00) | +9.76\% (0.90)  | +11.84\% (0.85) |
>     | **w/o** MR | +64.57\% (2.09) | +56.31\% (1.61) | +45.65\% (1.34) | +34.09\% (1.18) | +30.49\% (1.07) | +32.89\% (1.01) |

---

> ### Author Response · Authors · 2025-11-21
>
> **Q3 External Generalization on Public Datasets.**
>
> Thank you for raising this important question. We fully agree that validation on public datasets is crucial for demonstrating generalizability and transferability.
>
> - **Existing Evidence of Generalization:**
>
> We have already demonstrated in the paper that AEMP learns features with strong internal transferability and robustness to environmental variations through Cross-Region and Cross-Device experiments. In addition, we include new Cross-User experiments. The results are summarized in the following table and show that the proposed pre-training method effectively extracts user-independent localization features, which is a key factor in assessing its practical utility in real-world scenarios.
>
> |                       | 1           | 2           | 3           | 4           | 5           | 6           | 7           | 8           | 9           | 10          | 11          |
> |-----------------------|-------------|-------------|-------------|-------------|-------------|-------------|-------------|-------------|-------------|-------------|-------------|
> | **Height / Weight**   | 178 / 68    | 172 / 63    | 178 / 59    | 178 / 52    | 177 / 67    | 170 / 50    | 173 / 60    | 175 / 68    | 173 / 65    | 183 / 73    | 176 /76     |
> | **50th / 90th Error** | 0.65 / 1.38 | 0.72 / 1.65 | 0.88 / 3.31 | 0.91 / 3.47 | 0.73 / 1.88 | 0.75 / 1.81 | 0.78 / 1.88 | 0.92 / 3.03 | 0.90 / 1.93 | 0.85 / 2.64 | 0.85 / 1.97 |
>
> - **Public Dataset Validation**
>
> We conduct experiments on the available WILD-v2 dataset published by DLoc \citep{dloc}. The dataset consists of CSI samples from two visually similar environments, Env-1 and Env-2. During testing, 4000 samples from Env-1 and 1000 samples from Env-2 are used as the test set. Pretraining and fine-tuning are performed using only Env-1.
>
> | **Model**        | 10\% (m)       | 20\% (m)        | 40\% (m)        | 60\% (m)        | 80\% (m)        | 100\% (m)       |
> |------------------|----------------|-----------------|-----------------|-----------------|-----------------|-----------------|
> | RFM              | 2.34 / 5.22    | 1.92 / 4.29     | 1.50 / 3.34     | 1.39 / 3.14     | 1.38 / 3.05     | 1.31 / 2.89     |
> | Wireless-SSL     | 2.28 / 4.77    | 2.10 / 4.34     | 1.86 / 3.92     | 1.89 / 3.80     | 1.85 / 3.71     | 1.80 / 3.68     |
> | LocalGPT         | 1.52 / 3.29    | 1.44 / 3.16     | 1.31 / 2.91     | 1.29 / 2.91     | 1.20 / 2.83     | 1.22 / 2.82     |
> | Glow             | 1.50 / 2.92    | 1.32 / 2.82     | 1.36 / 2.96     | 1.33 / 2.86     | 1.25 / 2.85     | 1.24 / 2.86     |
> | Bert-WiFi        | 3.07 / 5.32    | 3.08 / 5.39     | 3.12 / 5.42     | 3.09 / 5.37     | 3.03 / 5.31     | 3.04 / 5.29     |
> | **AEMP**         |**1.08 / 2.35** | **1.02 / 2.24** | **0.95 / 2.06** | **1.00 / 2.25** | **0.96 / 2.19** | **0.92 / 2.12** |
>
> The results show that most existing methods perform poorly on the public dataset, mainly due to insufficient pretraining data collected within a single day, failing to capture time-invariant and generalizable channel features. In contrast, the proposed method suppresses environment-specific interference and focuses on inherent location-dependent features, achieving more robust positioning performance, with a median error of only 0.92m, even with limited single-day pretraining data.
>
> ---
>
> ### **Add Relevant References**
>
> [1] Xin Li, Hongbo Wang, Zhe Chen, Zhiping Jiang, and Jun Luo, "Uwb-fi: Pushing wi-fi towards ultra-wideband for fine-granularity sensing," in *Proceedings of the 22nd Annual International Conference on Mobile Systems, Applications and Services*, doi: 10.1145/3643832.3661889.
>
> [2] Yan Gao, Hongbo Xu, Chao Li, Yong Wang, and Zhiguo Shi, "Passive human sensing based on uwb cir," in *IEEE/CIC International Conference on Communications in China*, doi: 10.1109/ICCC62479.2024.10681796.
>
> [3] Xiaofeng Zhong, Yinfeng Xiang, Fang Yi, Chao Li, and Qinmin Yang, "Hmp-llm: Human mobility prediction based on pre-trained large language models," in *IEEE International Conference on Digital Twins and Parallel Intelligence*, doi: 10.1109/DTPI61353.2024.10778764.

---

### Official Review · Reviewer_d1Ze · 2025-10-31

**Soundness:** 2
**Presentation:** 2
**Contribution:** 2
**Rating:** 6
**Confidence:** 3

**Summary:**

This paper proposes a self-supervised pre-training mechanism for CSI-based data-driven indoor localization systems that aims to capture the spatiotemporal dependencies of wireless signals, providing robust representations that enhance the robustness and generalization capabilities of downstream indoor localization systems. The Spatial component is learned by training the model to perform 2D mask reconstruction as well as the model's confidence in the mask prediction, represented by the predicted variance, while temporal consistency is taught to the model by auto-regressively predicting the future frames. Moreover, the authors collect a large-scale dataset of CSI measurements from a large number of APs coming from a diverse set of mobile phones, and from different geographical locations over several days. Several studies and evaluations are performed to quantify the impact of the two proposed sub-components, as well as some experiments to quantify the proposed approach's capability to make the downstream systems robust to varying deployment settings. Moreover, quantitative analysis shows that the proposed approach outperforms other pretraining methods in terms of median and tail localization error.

**Strengths:**

- The proposed work maintains both spatial and temporal properties of wireless signal propagation
- The proposed methodology is well motivated and decently evaluated.

**Weaknesses:**

- The work does not fully evaluate the impact of the different proposed components on the mentioned challenges. For example, it is not clear the contribution of the Masked Reconstruction (MR) on generalizing to new areas or across different devices
- The work is only evaluated on the constructed dataset, and it is not clear how well the performance gains from the proposed pre-training methods carry over to other well-established datasets.

Overall, this proposed work can be a significant contribution in enhancing indoor localization systems, but would benefit from some additional clarifications and experiments. Based on these clarifications and experiments from the authors, I would be willing to revise this score.

Clarifying the following points in writing or by performing minor additional experiments would help quantify the effects and impacts of several design choices:
- Which part of the dataset was used as a validation split?
- What is the impact of multi-view fusion in the MR component on the generalization capability of downstream models? What is the impact of an increasing number of combinations by dropping more than 1 AP?
- I think the fact that almost all the baselines are underperforming compared to systems without pretraining is counterintuitive and warrants discussing and possibly evaluating.
- Since the position of the APs are encoded in the input, can a model trained on one test area be directly evaluated on another? I believe including the performance without additional fine-tuning in Table 4 would be a good addition.
- It would be interesting to see how reproducible the results are on other existing and well-established datasets, and to show that the proposed approach is not over-fitting on some nuanced features of the environments where the data was collected. This would help strengthen the proposed approach, as well as validate the proposed dataset.
- What is the structure of the MLP used as a task-specific head?


Minor comments:
- In lines 216 and 217, the N-1 and N should be swapped.

**Questions:**

Please focus on the weakenesses.

---

> ### Author Response · Authors · 2025-11-23
>
> We sincerely thank you for the valuable comments and suggestions.
>
> ---
>
> **Q1 Which part of the dataset was used as a validation split?**
>
> We use 20% of the fine-tuning data from each downstream task as a validation set. This 20% is selected randomly but in a stratified manner to ensure a balanced distribution across different collection dates. It is important to note that our pre-training/fine-tuning data and test data come from different time periods. The pre-training and fine-tuning datasets are collected in various regions during July and contain one week of dynamic CSI measurements. In contrast, the test datasets are collected in different regions during May and June, including 21 consecutive days of data from region 4A and 31 consecutive days from regions 5B and 5C. Because the training and test datasets are entirely disjoint, there is no risk of data leakage.
>
> ---
>
> **Q2 What is the impact of multi-view fusion in the MR component on the generalization capability of downstream models? What is the impact of an increasing number of combinations by dropping more than 1 AP?**
>
> We acknowledge the importance of these design choices for enhancing model robustness. We introduce multi-view combination in the paper to reduce the model's dependence on specific AP signals and thereby improve robustness to changes in AP configurations. To validate the effectiveness of multi-view combination, we conduct ablation studies on this component. We first replace the span mask (SM) in the w/o-AP model with random masking, denoted as w/o SM. Building on this variant, we then remove multi-view combination (MVC), denoted as w/o MVC. As summarized in the following table, removing MVC results in substantial degradation in downstream localization performance.
>
> |             | 10\% (m)        | 20\% (m)        | 40\% (m)        | 60\% (m)        | 80\% (m)        | 100\% (m)       |
> |-------------|-----------------|-----------------|-----------------|-----------------|-----------------|-----------------|
> | **AEMP**    | **1.27**        | **1.03**        | **0.92**        | **0.88**        | **0.82**        | **0.76**        |
> | **w/o** AP  | +7.87\% (1.37)  | +10.68\% (1.14) | +8.70\% (1.00)  | +13.64\% (1.00) | +9.76\% (0.90)  | +11.84\% (0.85) |
> | **w/o** SM  | +39.37\% (1.77) | +40.78\% (1.45) | +33.70\% (1.23) | +32.95\% (1.17) | +36.59\% (1.12) | +43.42\% (1.09) |
> | **w/o** MVC | +41.73\% (1.80) | +47.57\% (1.52) | +51.09\% (1.39) | +55.68\% (1.37) | +56.10\% (1.28) | +55.26\% (1.18) |
>
> We evaluate the effect of discarding different numbers of APs under the setting where the model is fine-tuned with only one day of labeled data, in order to assess the robustness of the localization task under limited supervision. For each frame, we retain the 6 APs with the strongest RSSI. As shown in the following table, the localization performance degrades progressively as more APs are discarded. This occurs because removing additional APs increases the number of multi-view combinations (except for the final group), while simultaneously reducing the number of APs included in each individual view. Consequently, the amount of AP information available for integration decreases, and since localization fundamentally relies on combining features extracted from multiple APs to infer spatial coordinates, the reduced AP coverage directly harms performance. Taken together, these results indicate that discarding exactly one AP represents the optimal configuration for implementing this data-augmentation strategy.
>
> | **Discarded APs** | **4A (m)**  | **5B5C (m)** | **Dimension** |
> |-------------------|-------------|--------------|---------------|
> | 1 AP              | 1.04 / 2.72 | 1.35 / 3.25  | 6x5           |
> | 2 AP              | 1.16 / 3.05 | 1.48 / 4.54  | 15x4          |
> | 3 AP              | 1.18 / 3.12 | 1.46 / 4.11  | 20x3          |
> | 4 AP              | 1.54 / 3.94 | 1.75 / 5.17  | 15x2          |
> | 5 AP              | 3.44 / 8.55 | 3.56 / 11.07 | 6x1           |

---

> > ### Author Response · Authors · 2025-11-23
> >
> > **Q3 I think the fact that almost all the baselines are underperforming compared to systems without pretraining is counterintuitive and warrants discussing and possibly evaluating.**
> >
> > We fully agree that this phenomenon appears counter-intuitive. The underlying reason lies in the nature of our dataset: it is large-scale, highly dynamic, and collected in real-world environments (as described in the dataset section). This setting poses substantial challenges for existing baselines. Methods such as Bert-WiFi and Wireless-SSL are primarily designed for relatively small and static CSI datasets. Their pre-training objectives do not incorporate the mechanisms that are essential for handling our data, including uncertainty-aware reconstruction through predicted variance in the masked reconstruction component and long-term temporal consistency constraints in the autoregressive prediction component. Consequently, these baselines struggle to cope with the high noise levels and spatio-temporal instability present in our dataset, leading to degraded localization performance even relative to supervised models trained with extensive labeled data. The strong performance of AEMP demonstrates that its customized objectives are well aligned with the intrinsic challenges of dynamic CSI environments.
> >
> > It is also important to note that, although supervised models outperform most pre-trained methods in long-term evaluation, this advantage needs access to large amounts of labeled data. As shown in the following table, when the proportion of labels is reduced to 50%, supervised performance deteriorates rapidly. In contrast, the pre-trained systems maintain stable localization accuracy under limited supervision and do not exhibit significant error inflation.      This highlights the dependence of supervised approaches on dense annotation and underscores the practical value of AEMP in large-scale indoor scenarios where labeled data collection is costly.
> >
> > | **Method**       | 10\% (m)        | 20\% (m)        | 40\% (m)        | 60\% (m)        | 80\% (m)        | 100\% (m)       |
> > |------------------|-----------------|-----------------|-----------------|-----------------|-----------------|-----------------|
> > | RFM              | 1.54 / 3.92     | 1.37 / 3.61     | 1.32 / 3.56     | 1.30 / 3.56     | 1.30 / 3.56     | 1.30 / 3.53     |
> > | Wireless-SSL     | 1.52 / 3.63     | 1.40 / 3.58     | 1.39 / 3.53     | 1.36 / 3.48     | 1.34 / 3.38     | 1.30 / 3.40     |
> > | LocalGPT         | 2.70 / 7.54     | 2.25 / 6.17     | 1.94 / 5.66     | 1.90 / 5.09     | 1.88 / 5.17     | 1.75 / 4.79     |
> > | Glow             | 1.45 / 3.86     | 1.41 / 3.78     | 1.37 / 3.70     | 1.38 / 3.67     | 1.36 / 3.70     | 1.36 / 3.71     |
> > | Bert-WiFi        | 1.28 / 3.31     | 1.10 / 3.04     | 1.07 / 2.75     | 0.99 / 2.62     | 1.01 / 2.64     | 0.96 / 2.56     |
> > | w/o pre-training | 2.00 / 5.00     | 1.63 / 4.09     | 1.50 / 3.83     | 1.20 / 3.04     | 1.08 / 2.81     | 1.06 / 2.69     |
> > | **AEMP**         | **1.27 / 3.30** | **1.03 / 3.03** | **0.92 / 2.65** | **0.88 / 2.45** | **0.82 / 2.32** | **0.76 / 2.27** |

---

> ### Author Response · Authors · 2025-11-23
>
> **Q4 Since the position of the APs are encoded in the input, can a model trained on one test area be directly evaluated on another? I believe including the performance without additional fine-tuning in Table 4 would be a good addition.**
>
> We appreciate the suggestion that prompted us to conduct this critical zero-shot transfer experiment.   We pre-train and fine-tune the model using data from the 2A area and then test it directly on the 4A and 5B5C datasets.   To prevent overfitting to 2A’s environmental characteristics, we freeze all components except the downstream task head.   The results, summarized in the table, show a significant degradation in localization accuracy for the w/o fine-tune model compared to those fine-tuned on the target test areas.
>
> |                     | **10\% (m)** | **20\% (m)** | **40\% (m)** | **60\% (m)** | **80\% (m)** | **100\% (m)** |
> |---------------------|--------------|--------------|--------------|--------------|--------------|---------------|
> | 4A (finetune)       | 1.95 / 5.10  | 1.39 / 3.85  | 1.16 / 3.28  | 1.08 / 3.00  | 0.98 / 2.78  | 0.92 / 2.69   |
> | 4A (w/o finetune)   | 3.19 / 8.75  | 3.34/ 9.25   | 3.17 / 8.97  | 3.21 / 8.92  | 3.26 / 9.13  | 3.24 / 9.08   |
> | 5B5C (finetune)     | 3.66 / 10.86 | 2.24 / 6.79  | 1.48 / 4.42  | 1.25 / 3.37  | 1.21 / 3.20  | 1.09 / 2.89   |
> | 5B5C (w/o finetune) | 6.38 / 17.43 | 6.68 / 19.56 | 6.44 / 17.60 | 6.53 / 18.24 | 6.59 / 18.93 | 6.88 / 20.25  |
>
> The failure of zero-shot transfer stems from environment-specific coupling.   Each region has a unique layout, with distinct AP coordinate distributions and occlusion areas that influence the CSI signal in ways not only simple AP coordinate changes.   During fine-tuning in 2A, the regression head learns a specialized mapping.   While the frozen encoder remains general, the uncalibrated regression head struggles to map learned embeddings to the new coordinates in other regions, causing a sharp performance drop.
>
> We acknowledge that perfect zero-shot transfer is unrealistic in CSI-based localization due to such strong environmental coupling. The true value of AEMP does not lie in eliminating all fine-tuning but in learning features that are significantly more general and easier to adapt than those of existing baselines. Consequently, in a new target area, the model requires only a minimal amount of target-domain data to quickly calibrate the regression head to the new propagation environment, achieving a notable improvement over baseline methods in final performance. As shown in the following table, compared with fully supervised models, AEMP attains superior accuracy even under limited labeled data.
>
> |                         | **10\% (m)** | **20\% (m)** | **40\% (m)** | **60\% (m)** | **80\% (m)** | **100\% (m)** |
> |-------------------------|--------------|--------------|--------------|--------------|--------------|---------------|
> | 4A                      | 1.95 / 5.10  | 1.39 / 3.85  | 1.16 / 3.28  | 1.08 / 3.00  | 0.98 / 2.78  | 0.92 / 2.69   |
> | 4A (w/o pre-training)   | 2.00 / 5.00  | 1.63 / 4.09  | 1.50 / 3.83  | 1.20 / 3.04  | 1.08 / 2.81  | 1.06 / 2.69   |
> | 5B5C                    | 3.66 / 10.86 | 2.24 / 6.79  | 1.48 / 4.42  | 1.25 / 3.37  | 1.21 / 3.20  | 1.09 / 2.89   |
> | 5B5C (w/o pre-training) | 4.28 / 10.56 | 3.66 / 9.17  | 2.26 / 5.36  | 1.94 / 4.69  | 1.55 / 3.55  | 1.38 / 3.30   |

---

> ### Author Response · Authors · 2025-11-23
>
> **Q5 It would be interesting to see how reproducible the results are on other existing and well-established datasets, and to show that the proposed approach is not over-fitting on some nuanced features of the environments where the data was collected. This would help strengthen the proposed approach, as well as validate the proposed dataset.**
>
> We accept this suggestion and agree that validating the model's transferability and robustness on external, publicly available, well-established datasets is essential for strengthening the contribution of the paper.
>
> We conduct experiments on the available WILD-v2 dataset published by DLoc \citep{dloc}. The dataset consists of CSI samples from two visually similar environments, Env-1 and Env-2. During testing, 4000 samples from Env-1 and 1000 samples from Env-2 are used as the test set. Pretraining and fine-tuning are performed using only Env-1.
>
> | **Model**        | 10\% (m)       | 20\% (m)        | 40\% (m)        | 60\% (m)        | 80\% (m)        | 100\% (m)       |
> |------------------|----------------|-----------------|-----------------|-----------------|-----------------|-----------------|
> | RFM              | 2.34 / 5.22    | 1.92 / 4.29     | 1.50 / 3.34     | 1.39 / 3.14     | 1.38 / 3.05     | 1.31 / 2.89     |
> | Wireless-SSL     | 2.28 / 4.77    | 2.10 / 4.34     | 1.86 / 3.92     | 1.89 / 3.80     | 1.85 / 3.71     | 1.80 / 3.68     |
> | LocalGPT         | 1.52 / 3.29    | 1.44 / 3.16     | 1.31 / 2.91     | 1.29 / 2.91     | 1.20 / 2.83     | 1.22 / 2.82     |
> | Glow             | 1.50 / 2.92    | 1.32 / 2.82     | 1.36 / 2.96     | 1.33 / 2.86     | 1.25 / 2.85     | 1.24 / 2.86     |
> | Bert-WiFi        | 3.07 / 5.32    | 3.08 / 5.39     | 3.12 / 5.42     | 3.09 / 5.37     | 3.03 / 5.31     | 3.04 / 5.29     |
> | **AEMP**         |**1.08 / 2.35** | **1.02 / 2.24** | **0.95 / 2.06** | **1.00 / 2.25** | **0.96 / 2.19** | **0.92 / 2.12** |
>
> The results show that most existing methods perform poorly on the public dataset, mainly due to insufficient pretraining data collected within a single day, failing to capture time-invariant and generalizable channel features. In contrast, the proposed method suppresses environment-specific interference and focuses on inherent location-dependent features, achieving more robust positioning performance, with a median error of only 0.92m, even with limited single-day pretraining data.
>
> ---
>
> **Q6 What is the structure of the MLP used as a task-specific head?**
>
> The task-specific head used for downstream localization is a simple MLP consisting of a Linear layer followed by a LeakyReLU activation, a Dropout layer, and another Linear layer.
>
> ---
>
> **Q7 In lines 216 and 217, the N-1 and N should be swapped.**
>
> We appreciate the reviewer's careful identification of this textual error. We confirm that the issue appears in the paragraph describing the multi-view combination strategy, and we commit to correcting N-1 and N in lines 216 and 217 in the revised manuscript.
>
> ---
>
> ### **Add Relevant References**
>
> [1] Kang Yang, Yuning Chen, and Wan Du, "OrchLoc: In-Orchard Localization via a Single LoRa Gateway and Generative Diffusion Model-based Fingerprinting," in *Proceedings of the 22nd Annual International Conference on Mobile Systems, Applications and Services*, doi: 10.1145/3643832.3661876.
>
> [2] Mai Ibrahim, Marwan Torki, and Mustafa ElNainay, "Cnn based indoor localization using rss time-series," in *IEEE Symposium on Computers and Communications*, doi: 10.1109/ISCC.2018.8538530.
>
> [3] Moustafa Abbas, Moustafa Elhamshary, Hamada Rizk, Marwan Torki, and Moustafa Youssef, "Wideep: Wifi-based accurate and robust indoor localization system using deep learning," in *IEEE International Conference on Pervasive Computing and Communications*, doi: 10.1109/PERCOM.2019.8767421.

---

### Official Review · Reviewer_H4cu · 2025-11-04

**Soundness:** 3
**Presentation:** 2
**Contribution:** 2
**Rating:** 6
**Confidence:** 3

**Summary:**

This paper proposes AEMP (Autoregressive-Enhanced Masked Pre-training), a self-supervised framework designed to improve robustness and generalization in indoor localization using WiFi Channel State Information (CSI). Traditional masked modeling fails to capture the spatiotemporal dependencies of CSI signals and is sensitive to environmental variations. To address these issues, AEMP integrates a hierarchical Transformer architecture with two complementary pretext tasks: masked reconstruction for learning spatial features and autoregressive prediction for temporal consistency. The authors also introduce span masking and multi-view fusion strategies to enhance resilience to deployment changes. Extensive experiments on the proposed ISACLoc dataset show that AEMP achieves superior localization accuracy and generalization compared with state-of-the-art baselines.

**Strengths:**

(1) The first pre-training framework for wireless sensing that combines masked modeling with autoregressive prediction to improve temporal coherence in CSI representation learning.

(2) A well-designed hierarchical Transformer architecture that separates spatial and temporal modeling via parameter-shared subnetworks.

(3) Introduction of span masking and multi-view fusion to simulate real-world dynamics and improve robustness to varying access point configurations.

(4) Comprehensive experiments, including cross-region, cross-device, and low-label scenarios, showing consistent improvements in accuracy and stability. Clear ablation and fine-tuning analyses demonstrating the role of each module in performance gains.

**Weaknesses:**

(1) The abstract does not effectively highlight the research gap or the motivation. Readers must infer the challenges being addressed from the introduction.

(2) Figure 2 is difficult to interpret due to small and cluttered text; visualization clarity is limited.

(3) While the gap is well-defined, the proposed solution lacks strong novelty, primarily integrating known ideas (masked modeling and autoregression) with moderate architectural adjustments.

(4) Although the method section is detailed, the flow is dense, and the conceptual link between the two tasks (masked reconstruction and autoregression) could be better articulated.

(5) while thorough, results mostly compare against conventional pretraining baselines. Broader comparisons (e.g., contrastive or diffusion-based pretraining methods) would better support novelty claims.

(6) The ISACLoc dataset setup is interesting, but reproducibility could be improved by including open-source plans or quantitative data statistics.

**Questions:**

(1) The combination of Gaussian NLL loss for masking and weighted MSE for prediction is sensible, but how is the stability of training?

(2) How sensitive is AEMP to the weighting and scheduling parameters (e.g., λ and η) in Equation (9)? Were these tuned heuristically or via validation?

(3) Since span masking and multi-view fusion both modify spatial inputs, how do they interact? Could one suffice without the other?

(4) How scalable is AEMP to different wireless protocols or hardware with different CSI formats—does the pretraining transfer effectively?

---

> ### Author Response · Authors · 2025-11-21
>
> Thank you for your valuable comments and kind words to our work.
>
> ---
>
> **Q1 The abstract does not effectively highlight the research gap or the motivation. Readers must infer the challenges being addressed from the introduction.**
>
> We apologize for the lack of clarity in the abstract. The abstract is intended to highlight the difficulty of obtaining high-quality, large-scale labeled datasets for learning-based CSI indoor localization and to point out the limitations of existing masking-based pre-training methods in dynamic and large-scale CSI scenarios. In the revised version, we update the abstract to state the research gap and motivation more directly and concisely in the opening sentences. Here is our revised abstract:
>
> The major obstacle for learning-based Channel State Information (CSI) localization is to obtain a high-quality large-scale annotated dataset. However, unlike visual datasets that can be easily annotated by human workers, CSI signals are  RF signal is non-intuitive and non-interpretable, making the annotation process both time-consuming and labor-intensive. Considering the potential of self-supervised learning to reduce reliance on labeled data, masked reconstruction has emerged as a promising alternative. However, directly applying existing designs to large-scale CSI scenarios faces unique challenges, including unstable representations in unmasked regions, inability to preserve long-range channel correlations, and high sensitivity to variations in access point layouts and propagation environments. These issues significantly degrade localization performance in downstream tasks. To address these issues, we propose an autoregressive-enhanced masked pre-training (AEMP) framework. AEMP employs a hierarchical Transformer architecture where spatial subnetworks perform masked reconstruction to capture local channel features, while a temporal network enforces consistency through autoregressive prediction. In addition, multi-view fusion and span masking improve robustness under dynamic deployment conditions. Extensive experiments demonstrate that AEMP yields stable and transferable representations, achieving superior performance and strong generalization on downstream indoor localization tasks. To the best of our knowledge, this is the first pre-training framework for wireless sensing that integrates temporal prediction to complement masked reconstruction.
>
> ---
>
> **Q2 Figure 2 is difficult to interpret due to small and cluttered text; visualization clarity is limited.**
>
> We agree with this observation.  We revise Figure 2 substantially to enhance visual clarity, enlarge the font, and simplify the workflow, ensuring that all major figures remain easy to understand in the updated version.

---

> > ### Author Response · Authors · 2025-11-21
> >
> > **Q3 While the gap is well-defined, the proposed solution lacks strong novelty, primarily integrating known ideas (masked modeling and autoregression) with moderate architectural adjustments.**
> >
> > We sincerely appreciate your insightful questions regarding the novelty and necessity of integrating Masked Reconstruction (MR) with Autoregressive Prediction (AP). Your comments directly touch upon the theoretical foundation of the AEMP architecture and motivate us to provide a more rigorous and comprehensive explanation.
> >
> > We respectfully argue that the integration of MR and AP is a non-trivial and essential solution tailored to the high-dimensional spatial structure and strong temporal dependency inherent in CSI data.    Our innovation does not lie in merely stacking existing techniques but in designing a synergistic mechanism with cross-task regularization.
> >
> > - **Theoretical Motivation:**
> >
> > Applying traditional self-supervised methods directly to CSI data introduces fundamental limitations. While MR is effective at learning the instantaneous spatial geometry of the channel, it faces inherent challenges when applied to sequential data: the representations of unmasked tokens lack explicit supervision, often resulting in unstable embeddings that fail to capture long-term signal evolution. In CSI scenarios, this manifests as poor robustness to channel drift.
> >
> > In contrast, AP captures temporal dependencies but does not enforce the spatial reconstruction constraints needed to extract high-fidelity spatial semantics from noisy CSI inputs—semantics that are crucial for accurate indoor localization. Our key insight is that temporal constraints provided by AP must be used to stabilize and enhance the spatial features learned through MR.
> >
> > - **Core Innovation:**
> >
> > The novelty of AEMP lies in tightly coupling MR and AP through a hierarchical Transformer architecture. A shared spatial subnetwork learns robust instantaneous CSI embeddings via MR, while a temporal subnetwork performs sequence prediction on the reconstructed CSI. The temporal loss from AP backpropagates into the spatial encoder as a regularization signal, enforcing physical consistency of the reconstructed features across time. This directly addresses the instability of conventional MR on sequential data, allowing the learned representations to be not only spatially robust but also intrinsically aligned with long-term physical dynamics, which is reflected in the superior long-term localization performance (see Appendix A.2).
> >
> > This MR–AP integration is therefore not a simple combination of tasks but a time-enhanced masking paradigm in which AP acts as a temporal supervisor that stabilizes and refines the spatial features extracted by MR. The result is a representation that offers unprecedented robustness and generalization for downstream localization.
> >
> > Our ablation results clearly demonstrate the indispensability of both tasks: removing either component leads to notable performance degradation.  As shown in the following table, the localization error increases by 0.09 m w/o AP and by 0.25 m w/o MR, providing strong empirical evidence for the necessity of their joint optimization.
> >
> > |            | 10\% (m)        | 20\% (m)        | 40\% (m)        | 60\% (m)        | 80\% (m)        | 100\% (m)       |
> > |------------|-----------------|-----------------|-----------------|-----------------|-----------------|-----------------|
> > | **AEMP**   | **1.27**        | **1.03**        | **0.92**        | **0.88**        | **0.82**        | **0.76**        |
> > | **w/o** AP | +7.87\% (1.37)  | +10.68\% (1.14) | +8.70\% (1.00)  | +13.64\% (1.00) | +9.76\% (0.90)  | +11.84\% (0.85) |
> > | **w/o** MR | +64.57\% (2.09) | +56.31\% (1.61) | +45.65\% (1.34) | +34.09\% (1.18) | +30.49\% (1.07) | +32.89\% (1.01) |
> >
> > ---
> >
> > **Q4 Although the method section is detailed, the flow is dense, and the conceptual link between the two tasks (masked reconstruction and autoregression) could be better articulated.**
> >
> > We fully agree with this suggestion. We revise Sections 3.4–3.6 (multi-task joint learning, mask reconstruction, and autoregressive prediction) extensively to more clearly emphasize the error propagation mechanism and to explain how autoregressive prediction enforces global temporal consistency on the spatial features learned through mask reconstruction, thereby improving the workflow in this part.

---

> ### Author Response · Authors · 2025-11-21
>
> **Q5 while thorough, results mostly compare against conventional pretraining baselines. Broader comparisons (e.g., contrastive or diffusion-based pretraining methods) would better support novelty claims.**
>
> We appreciate this suggestion, which helps strengthen the validation. We initially focus on masking-based methods as the most direct competitors, such as Bert-WiFi and RFM. In the revised experimental section, we commit to adding comparisons with diffusion model–based pre-training methods. (Contrastive learning–based approaches are already included in existing baselines, and Glow combines graph neural networks with contrastive learning to perform pre-training tasks.) This further supports the effectiveness of AEMP.
>
> We design a pre-training method based on a denoising diffusion probabilistic model, denoted as DDPMLoc. Its average median error across different regions and localization performance under fine-tuning with varying label proportions are presented in the table below.
>
> | **Method**  | **4A (m)**  | **5B5C (m)**| **Mean (m)**|
> |-------------|-------------|-------------|-------------|
> | Glow        | 1.36 / 3.71 | 1.52 / 4.93 | 1.44 / 4.32 |
> | DDPMLoc     | 1.21 / 3.41 | 1.37 / 4.43 | 1.29 / 3.92 |
> | **AEMP**    | 0.76 / 2.27 | 1.03 / 3.03 | 0.90 / 2.65 |
>
>
> | **Method**  | 10\% (m)        | 20\% (m)        | 40\% (m)        | 60\% (m)        | 80\% (m)        | 100\% (m)       |
> |-------------|-----------------|-----------------|-----------------|-----------------|-----------------|-----------------|
> | Glow        | 1.45 / 3.86     | 1.41 / 3.78     | 1.37 / 3.70     | 1.38 / 3.67     | 1.36 / 3.70     | 1.36 / 3.71     |
> | DDPMLoc     | 1.32 / 3.52     | 1.30 / 3.50     | 1.25 / 3.51     | 1.24 / 3.47     | 1.22 / 3.45     | 1.21 / 3.41     |
> | **AEMP**    | **1.27 / 3.30** | **1.03 / 3.03** | **0.92 / 2.65** | **0.88 / 2.45** | **0.82 / 2.32** | **0.76 / 2.27** |
>
> ---
>
> **Q6 The ISACLoc dataset setup is interesting, but reproducibility could be improved by including open-source plans or quantitative data statistics.**
>
> We appreciate your emphasis on this important issue. We fully recognize that dataset accessibility is crucial for ensuring the reproducibility of indoor localization research based on large-scale commercial WiFi systems and for advancing the field. We are pleased to inform the community that we are actively preparing the public release of the ISACLoc dataset, which is scheduled within the next three months. The release will include not only the dataset itself but also comprehensive documentation and usage guidelines to facilitate adoption by the research community.

---

> ### Author Response · Authors · 2025-11-21
>
> **Q7 The combination of Gaussian NLL loss for masking and weighted MSE for prediction is sensible, but how is the stability of training?**
>
> We sincerely appreciate your recognition of the soundness of our loss function design. The training process of the AEMP framework exhibits remarkable stability and rapid convergence, which primarily stems from our explicit modeling of CSI uncertainty within the loss function and our progressively coupled pre-training strategy.
>
> For the masked reconstruction task, we employ the Gaussian NLL loss, as shown in Eq. 5 of the paper. This loss models an uncertainty-aware reconstruction objective by jointly predicting the mean and the log-variance. Given the highly dynamic and noisy nature of CSI signals, this mechanism enables the model to adaptively increase the predicted variance in challenging or NLoS regions. This prevents the severe gradient oscillations that traditional L2 loss often suffers when encountering outliers, thereby ensuring stable pre-training on large-scale and high-dynamic CSI datasets. It is worth noting that if the model becomes overly confident in its predictions, the log-variance may take negative values, which can adversely affect the masking loss. To avoid negative-loss issues, we clip the predicted log-variance to the range[−2,5], ensuring stable gradient descent throughout pre-training.
>
> Because AEMP integrates two self-supervised tasks, we adopt a progressively coupled (PC) strategy to maintain smooth and stable optimization. During the early stage of training, we stop the gradient flow from the autoregressive prediction branch. As the masked reconstruction task becomes stable, we enable full gradient propagation to achieve end-to-end pre-training. This progressive coupling prevents early-stage masked reconstruction from compromising its spatial modeling capability in order to satisfy the autoregressive objective, while still allowing temporal consistency to be introduced once the representations stabilize. As shown in the table, the progressive coupling strategy significantly outperforms both the fully end-to-end (E2E) and the stop-gradient (SG) alternatives under limited supervision, reflecting the strong stability of our training process.
>
>
> |**Mode**| 10\% (m)        | 20\% (m)        | 40\% (m)        | 60\% (m)        | 80\% (m)        | 100\% (m)        |
> |--------|-----------------|-----------------|-----------------|-----------------|-----------------|------------------|
> | E2E    | 1.55 / 3.99     | 1.14 / 3.21     | 0.97 / 2.72     | 0.90 / 2.57     | 0.87 / 2.48     | 0.84 / 2.36      |
> | SG     | 1.58 / 4.25     | 1.13 / 3.19     | 1.03 / 2.82     | 0.88 / 2.56     | 0.85 / 2.42     | 0.82 / 2.43      |
> | **PC** | **1.27 / 3.30** | **1.03 / 3.03** | **0.92 / 2.65** | **0.88 / 2.45** | **0.82 / 2.32** | **0.76 / 2.27**  |
>
> ---
>
> **Q8  How sensitive is AEMP to the weighting and scheduling parameters (e.g., $\lambda$ and $\eta$) in Equation (9)? Were these tuned heuristically or via validation?**
>
> Thank you for the thoughtful question. Our task weight $\lambda$ is neither heuristic nor determined through validation. Instead, we employ a cosine scheduler that anneals $\lambda$ smoothly from its initial value to its target value (from 0 to 1) over the course of training. This scheduling strategy regulates error propagation and prevents instability in the early stages of optimization. Only after the masked reconstruction objective becomes stable do we gradually increase the contribution of the autoregressive loss by raising $\lambda$, enabling the model to learn temporal consistency. This dynamic adjustment also avoids the suboptimal solutions that fixed loss weights may cause, thereby improving the robustness of optimization.
>
> The parameter $\eta$ controls the balance between the single-step prediction loss and the multi-step rollout loss within the autoregressive objective. Proper tuning of $\eta$ ensures that the model captures local temporal dependencies while maintaining the ability to predict long-range trajectories. We determine the value of $\eta$ through validation-set search. As shown in the following table, performance on the downstream localization task remains stable across a wide range of $\eta$, demonstrating that AEMP is not sensitive to this parameter.
>
> | **$\eta$**     | **50th Error (m)** | **90th Error (m)** |
> |----------------|--------------------|--------------------|
> | 0.1            | 0.93               | 2.55               |
> | 0.3            | 0.78               | 2.31               |
> | 0.5            | 0.80               | 2.29               |
> | **0.7 (Ours)** | **0.76**           | **2.27**           |
> | 0.9            | 0.87               | 2.56               |

---

> > ### Author Response · Authors · 2025-11-21
> >
> > **Q9 Since span masking and multi-view fusion both modify spatial inputs, how do they interact? Could one suffice without the other?**
> >
> > We appreciate your attention to the potential interaction between these two mechanisms. Span Masking (SM) and Multi-View Composition (MVC) address two orthogonal challenges in achieving robust CSI-based localization. They operate independently in both mechanism and objective, and neither can substitute for the other.
> >
> > Span Masking is designed to enhance robustness against instantaneous signal-quality variations such as data corruption or occlusion. It operates within the feature sequence of each individual AP view.
> > Multi-View Composition targets robustness in the spatial dimension. By randomly composing AP subsets during pre-training, it enforces the learning of AP-configuration-invariant representations, allowing the model to generalize across deployment-specific topologies.
> >
> > To demonstrate their complementarity, we conduct ablation studies on both components. We replace SM in the w/o-AP model with random masking, denoted as w/o SM. Moreover, to further examine the effectiveness of multi-view integration, we remove MVC on top of the w/o SM model, denoted as w/o MVC. As summarized in the following table, removing MVC or replacing SM leads to substantial degradation in downstream localization performance.
> >
> >
> > |             | 10\% (m)        | 20\% (m)        | 40\% (m)        | 60\% (m)        | 80\% (m)        | 100\% (m)       |
> > |-------------|-----------------|-----------------|-----------------|-----------------|-----------------|-----------------|
> > | **AEMP**    | **1.27**        | **1.03**        | **0.92**        | **0.88**        | **0.82**        | **0.76**        |
> > | **w/o** AP  | +7.87\% (1.37)  | +10.68\% (1.14) | +8.70\% (1.00)  | +13.64\% (1.00) | +9.76\% (0.90)  | +11.84\% (0.85) |
> > | **w/o** SM  | +39.37\% (1.77) | +40.78\% (1.45) | +33.70\% (1.23) | +32.95\% (1.17) | +36.59\% (1.12) | +43.42\% (1.09) |
> > | **w/o** MVC | +41.73\% (1.80) | +47.57\% (1.52) | +51.09\% (1.39) | +55.68\% (1.37) | +56.10\% (1.28) | +55.26\% (1.18) |

---

> ### Author Response · Authors · 2025-11-21
>
> **Q10 How scalable is AEMP to different wireless protocols or hardware with different CSI formats—does the pretraining transfer effectively?**
>
> We greatly appreciate your interest in the scalability of AEMP, which is one of the central design principles of our framework. AEMP exhibits strong scalability and transferability, primarily due to our input representation design. Instead of directly using raw CSI amplitude/phase measurements, our preprocessing module converts CSI samples into covariance matrices. As statistical descriptors that capture inter-antenna correlations, covariance matrices provide a high-level representation that is inherently insensitive to underlying physical-layer protocols and specific hardware parameters. This abstraction enables our Transformer encoder to process inputs from heterogeneous wireless standards or hardware platforms, as long as they can be mapped into this statistical feature space.
>
> Building on this representation, AEMP functions as a general-purpose sequential feature learner. As a result, the spatio-temporal priors acquired through our self-supervised objectives become domain-agnostic, allowing efficient transfer to new protocols or hardware configurations and significantly enhancing the practicality and scalability of the framework.
>
> To further validate the extensibility of AEMP, We conduct experiments on the available WILD-v2 dataset published by DLoc \citep{dloc}, whose CSI format is entirely different from that used in our study. The dataset consists of CSI samples from two visually similar environments, Env-1 and Env-2. During testing, 4000 samples from Env-1 and 1000 samples from Env-2 are used as the test set. Pretraining and fine-tuning are performed using only Env-1.
>
> | **Model**        | 10\% (m)       | 20\% (m)        | 40\% (m)        | 60\% (m)        | 80\% (m)        | 100\% (m)       |
> |------------------|----------------|-----------------|-----------------|-----------------|-----------------|-----------------|
> | RFM              | 2.34 / 5.22    | 1.92 / 4.29     | 1.50 / 3.34     | 1.39 / 3.14     | 1.38 / 3.05     | 1.31 / 2.89     |
> | Wireless-SSL     | 2.28 / 4.77    | 2.10 / 4.34     | 1.86 / 3.92     | 1.89 / 3.80     | 1.85 / 3.71     | 1.80 / 3.68     |
> | LocalGPT         | 1.52 / 3.29    | 1.44 / 3.16     | 1.31 / 2.91     | 1.29 / 2.91     | 1.20 / 2.83     | 1.22 / 2.82     |
> | Glow             | 1.50 / 2.92    | 1.32 / 2.82     | 1.36 / 2.96     | 1.33 / 2.86     | 1.25 / 2.85     | 1.24 / 2.86     |
> | Bert-WiFi        | 3.07 / 5.32    | 3.08 / 5.39     | 3.12 / 5.42     | 3.09 / 5.37     | 3.03 / 5.31     | 3.04 / 5.29     |
> | **AEMP**         |**1.08 / 2.35** | **1.02 / 2.24** | **0.95 / 2.06** | **1.00 / 2.25** | **0.96 / 2.19** | **0.92 / 2.12** |
>
> The results show that most existing methods perform poorly on the public dataset, mainly due to insufficient pretraining data collected within a single day, failing to capture time-invariant and generalizable channel features. In contrast, the proposed method suppresses environment-specific interference and focuses on inherent location-dependent features, achieving more robust positioning performance, with a median error of only 0.92m, even with limited single-day pretraining data.
>
> ---
>
> ### **Add Relevant References**
>
> [1] Wei Gong and Jiangchuan Liu, "Roarray: Towards more robust indoor localization using sparse recovery with commodity wifi," in *IEEE Transactions on Mobile Computing*, doi: 10.1109/TMC.2018.2860018.
>
> [2] Ho, Jonathan, Ajay Jain and P. Abbeel, "Denoising Diffusion Probabilistic Models," *arXiv preprint arXiv:2006.11239*, 2020.

---

### Author Response · Authors · 2025-11-25

Author Rebuttal

We sincerely thank all reviewers for their valuable comments and suggestions. We appreciate that the reviewers recognize our work as "well-designed", "well-motivated", demonstrating "strong generalization", and making "a significant contribution in enhancing indoor localization systems" with "decently evaluated". Each reviewer's feedback has been carefully addressed individually. The manuscript has been revised according to the suggestions provided. Here is a summary of our main revisions:

1. Model Analysis and Technical Details:
- Elucidated the architecture of the downstream task-specific MLP head.
- Provided ablation studies on span masking and multi-view combination components.
- Added a detailed analysis of AP-dropping strategies, showing the best performance when dropping one AP (50th error: 1.04 m).
- Analyzed the novelty and utility of the proposed AEMP.
- Provided a performance comparison with additional baselines:
  - AEMP: 0.76m 50th Error, 2.27m 90th Error
  - Diffusion-based DDPMLoc: 1.29m 50th Error, 3.92m 90th Error
  - Contrastive-learning-based Glow: 1.36m 50th Error, 3.71m 90th Error
- Discussed the reasons behind the weak performance of existing baseline methods.

2. Generalization and Cross-Domain Capability:
- Evaluated performance on the public WILD dataset:
  - AEMP: 0.92 m 50th Error, 2.12 m 90th Error
  - LocalGPT SOTA: 1.22 m 50th Error, 2.82 m 90th Error
- Verified cross-user generalization capability.
- Provided zero-shot results on unseen deployment areas without fine-tuning.

3. Dataset and Experimental Details:
- Explained the composition of the train/validation split (July data for training, May-June data for testing).
- Added an analysis of the loss weighting for the autoregressive prediction task during pretraining.
- Planned to release the ISACLoc dataset within three months.
- Provided a detailed description of all training losses.

4. Paper Writing:
- Revised the abstract to better highlight the research motivation.
- Redrew Figure 2 to make it easier to interpret.
- Rearticulated the conceptual connection between masked reconstruction and autoregressive prediction.

The revisions demonstrate AEMP's advantages in few-shot fine-tuning, accuracy, and generalization while addressing all major concerns. We look forward to further discussions and are happy to address any additional questions.

Best regards, Authors

---

### Note · Authors · 2026-01-15

I have read and agree with the venue's withdrawal policy on behalf of myself and my co-authors.